

# Flood hazard in Afghanistan is intensified both by natural and socioeconomic factors

Qutbudin Ishanch[1]*, Kanchan Mishra[1]*, Christiane Zarfl[1], Kathryn E. Fitzsimmons[1,2]

[1]Department of Geosciences, Eberhard Karls University of Tübingen, Schnarrenbergstraße 94-96, 72076 Tübingen, Germany
[2] School of Earth, Atmosphere and Environment, Monash University, 9 Rainforest Walk, Clayton VIC 3800, Australia

*Corresponding authors*: Qutbudin Ishanch (qutbudin.ishanch@sedgeo.uni-tuebingen.de); Kanchan Mishra (kanchan.mishra@uni-tuebingen.de)

**Abstract.**The increasing frequency of climate-driven extreme events, such as heavy precipitation, floods, and droughts, poses severe social, economic, and environmental challenges. Among these, floods are the most destructive, causing significant
damage to lives, property, and infrastructure. This study assesses flood risk in Afghanistan using remote sensing and geographic information systems to evaluate flood hazards and vulnerabilities at sub-basin and provincial levels. Principal Component Analysis (PCA) identifies key environmental, climatic, and social indicators influencing flood risk, while the Analytical Hierarchy Process (AHP) systematically ranks these indicators to ensure logical consistency and reduce bias in decision-making. Findings indicate that Afghanistan's eastern and northeastern regions, particularly within the Amu and Kabul
River basins, face the highest flood hazards due to precipitation, topography, and drainage characteristics that accelerate runoff. Vulnerability analysis highlights that densely populated rural areas in the northern and eastern regions are at greater risk, exacerbated by significant land use changes. This study provides critical insights for policymakers, identifying high-risk areas and supporting targeted resource allocation and mitigation strategies. The findings aim to enhance community preparedness and resilience against future flood risks.

## 1    Introduction

The rapid climatic changes over recent decades have raised global concern, particularly with respect to the impacts of associated extreme precipitation, floods and droughts. They have intensified in frequency, magnitude, and duration worldwide (Castellari and Kurnik, 2017; IPCC, 2007). Compared to other natural disasters, floods are considered to be more devastating in terms of their social, economic and environmental impacts  (Kron, 2005; de Moel et al., 2015). Over the last few decades,
numerous studies have reported the damage to life and infrastructure associated with flooding events (Abdel-Fattah et al., 2017; Kron, 2005; Ran and Nedovic-Budic, 2016): the October 2012 storm and associated flood which hit New York and New Jersey (United States of America (USA)) caused net damages exceeding US$ 60 billion (Aerts et al., 2013), while heavy floods in the Elbe and Danube basins of central Europe in June 2013 resulted in dozens of fatalities and a total loss of €12 billion (Schröter et al., 2015). In September 2022, Pakistan witnessed catastrophic flooding that submerged one-third of the country,
affected 33 million people, and caused economic losses worth US$ 30 billion (World Bank, 2022).





Flooding is defined as a natural phenomenon where a portion of land becomes temporarily inundated by excessive surface water, either from river channels or as the result of heavy precipitation (Kron, 2005; Nearing et al., 2024). The occurrence and magnitude of floods depends on factors such as topography, land use, soil type, and meteorological conditions (Klijn et al., 2015; Kron, 2002; de Moel et al., 2015). Rapid population growth and associated anthropogenic activities such as land-use

change, and hydraulic engineering are increasingly triggering more catastrophic flood impacts. Countries of the Global South are particularly vulnerable to floods due to less well developed infrastructure, mitigation, and institutional capacity (Hossain et al., 2020; Nearing et al., 2024; Poku-Boansi et al., 2020).

Flooding events cannot be prevented, but their destructive consequences can be minimised by assessing their risk and managing them accordingly (Kerkhoff et al., 2009; Kron, 2002; Rafiq and Blaschke, 2012). Therefore, flood risk mapping and assessment

is a critical first step on which region-specific appropriate management and mitigation strategies can be based to minimize the damage, loss and human suffering caused by flood events (Mishra and Sinha, 2020; de Moel et al., 2015). Flood risk assessment generates a clear and informative maps of flood-vulnerable areas to assist public institutions and decision-makers to implement mitigation strategies and prioritize resource allocation (Klijn et al., 2015; de Moel et al., 2015) in in a sustainable way (Price, 2006)."Flood risk" is a multidimensional concept derived from the product of hazard and vulnerability (Kron, 2002, 2005;

Lombana et al., 2024; Mishra and Sinha, 2020; de Moel et al., 2015; Pandey et al., 2010). *Hazard* refers to a physically threatening event, and its probability of occurrence within certain areas (Blaikie et al., 2014; Karmokar and De, 2020; Shehata and Mizunaga, 2018). *Vulnerability* assesses the values that might be at risk, such as life, property, or the lack of resources to recover from the consequences of flooding (Cutter, 2012; Müller et al., 2011; Schwarz and Kuleshov, 2022; Singh and Pandey, 2021). However, the indices defining both hazard and vulnerability varies across regions (Klijn et al., 2015); for instance,

physical and social systems disturbed by the same external force in different regions may respond differently depending on varying coping capacities and sensitivities (Membele et al., 2022; Nazeer and Bork, 2019; Van et al., 2022), making these interpretations directly relevant in the context of local studies.

Although significant efforts have been made to evaluate flood hazard, vulnerability and risk worldwide (Efraimidou and Spiliotis, 2024), political sensitivity, limited access and financial constrains add another layer of difficulty for assessing flood

risk in vulnerable places like  Afghanistan. This mountainous nation is not only prone to natural and climatic disasters (Hagen and Teufert, 2009; Qutbudin et al., 2019; Sediqi et al., 2022); decades of conflict, instability and environmental degradation have made it highly susceptible to crisis. Afghanistan ranks fourth as the country most at risk (Inform Risk Index, 2024), and seventh most vulnerable and least prepared for future climate change (Notre Dame Global Adaptation Index, 2024). The last decades alone have overseen multiple flooding events with catastrophic loss of life and damage to agricultural land, livestock,

and infrastructure (Hagen and Teufert, 2009). Landslides triggered by heavy rain in Badakhshan in the north of the country in 2014 killed 500 people and affected 27 provinces (OCHA, 2014). In August 2020, floods in Parwan near the capital of Kabul caused over 100 deaths and impacted more than 2,000 households (ECHO, 2020). Despite these staggering figures, limited studies been undertaken to assess flood sensitivity across Afghanistan beyond small-scale projects focussing on northern Kabul city (Manawi et al., 2020), Kabul River Basin (Iqbal et al., 2018; Tani and Tayfur, 2023) and Parwan province (Fazel-Rastgar





and Sivakumar, 2023). Given the vulnerability of this large and mountainous country, there is a clear need to map and assess flood sensitivity at the national scale.

Flood vulnerability can be assessed through regular monitoring, and is increasingly being undertaken using remote sensing and spatial data analysis. Regular monitoring of a large, poorly resourced country such as Afghanistan is not only time consuming but also logistically difficult and adds to its financial burden (Goyal et al., 2020). For data-scarce regions, remote

sensing (RS) and spatial data analysis with Geographic Information Systems (GIS) has emerged as an effective tool (Bhatt et al., 2014). A range of satellite derivatives and GIS techniques, many of which are open-source, are now available to assess entire country profiles for flood hazard and sensitivity (Holand et al., 2011; Membele et al., 2022; Mokhtari et al., 2023; Ogarekpe et al., 2020; Samela et al., 2018). This includes the recent development of remotely sensed hydro-morphological characterization over large spatial scales which overcomes the requirement for long-term hydrological data to incorporate into

traditional hydrological models (Teng et al., 2017).

The overall objective of this study is to map and assess flood risk in Afghanistan. Given the mountainous nature of the country and its administrative division into broadly hydro-morphically defined provinces, we approach this by assessing hazards at the subbasin. We identify the environmental and climatic parameters that drive risk in hydro-morphologically defined sub-basins using principal component analysis (PCA). We assess flood hazard and vulnerability, along with their driving indicators,

individually, and rank their relative importance using the Analytical Hierarchy Process (AHP) (Saaty, 1980). Our results highlight the most flood-vulnerable areas and are intended to serve as a basis for government and relief agencies to develop appropriate management plans. By incorporating social vulnerability into assessments for flood management, decision-makers can more effectively allocate resources to those communities most in need.

## 2   Study Site

The study investigates the entire nation of Afghanistan, a country in Central Asia with an area of around 652,000 sq.km (Tani and Tayfur, 2023) (Fig. 1a). It shares borders with Iran in the west, Pakistan in the east and south, China and Tajikistan in the northeast, Uzbekistan in the north and Turkmenistan in the northwest. Administratively, Afghanistan is divided into 34 provinces and 365 districts, which are broadly delineated according to river sub-basins (Najmuddin et al., 2022). The country hosts 29 major rivers with a combined length of approx. 35,000 km, spanning five major river basins: the (1) Panj - Amu, (2)

North, (3) Helmand, (4) Kabul and (5) Harirud Morghab (Qasimi et al., 2023; Shokory et al., 2023). The Hindu Kush mountains extend  from west to east (Tani and Tayfur, 2023), dividing the country into the central highlands which reach 8000 m asl, southwestern plateau and northern plains which have an average elevation of 150 m asl (Tani and Tayfur, 2023) (Figure. 1a). (Aich et al., 2017; Tani and Tayfur, 2023).

The strongly variable topography and associated climate, influences the landuse and landcover (LULC) of Afghanistan.

Approximately 81% (534,504 sq. km) of the country is dominated by barren land and sand cover; of the remainder, 12% of land is arable and or cultivated (69,914 sq. km) (Shrestha, 2007) (Table 1 and Fig. 1b). Urban or built-up areas comprise only



a small fraction (0.47% or 26,421 sq. km). Water bodies and marshlands occupy 2.85% (15,815 sq. km), and forest or shrublands cover 2.78% (16,605 sq. km). Permanent snow cover is least extensive and constitutes 0.76% (4,167 sq. km).


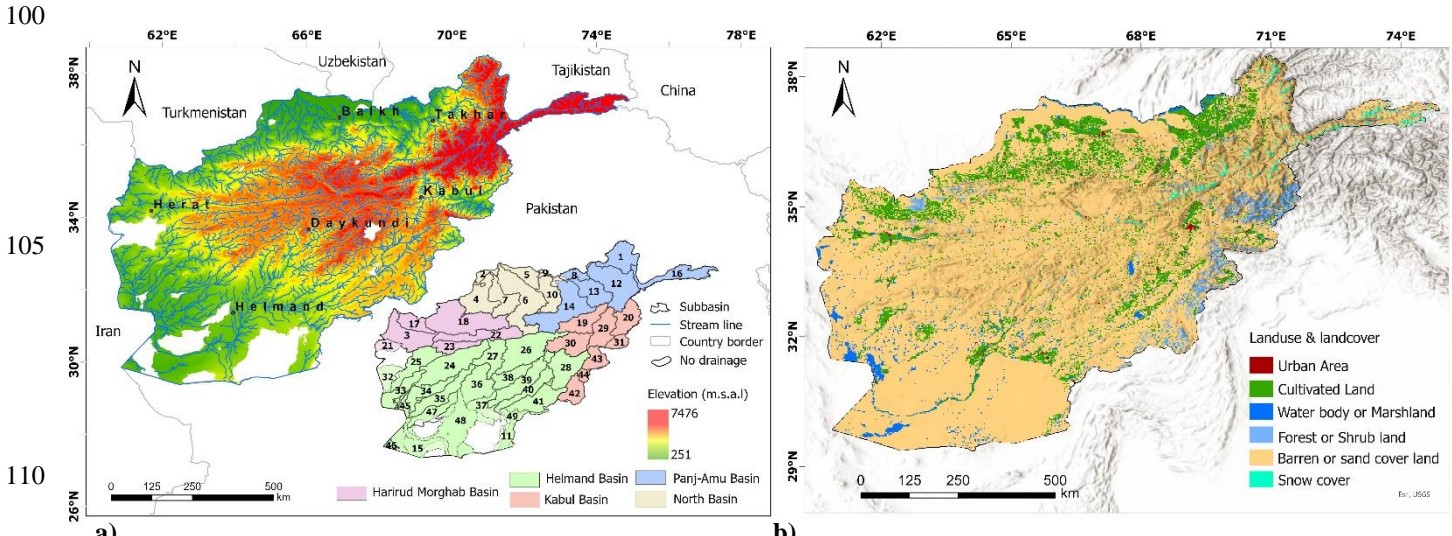



**Figure 1:** a) Geographical map showing the Afghanistan region with elevation, river lines with major cities/towns along with the subsidiary map with major basin boundaries and associated subbasin number; b) Map shows major landuse and landcover of the study region.

Afghanistan has a continental semi-arid through to arid climate, and incorporates desert, steppe and highland landscapes with temperature and precipitation patterns typical of these (Bronkhorst and Bhandari, 2021; Shokory et al., 2023). Temperatures are strongly continental, with hot summers and cold winters (Freitag et al., 2010), and variation with altitude: mountainous areas often remain below zero year-round, while southern arid regions frequently exceed 35°C (Bronkhorst and Bhandari, 2021).

**Table 1** Landuse and landcover areal statistics of Afghanistan

| Landuse and land cover class name | Area (sq.km) | Percentage of area covered |
|---|---|---|
| Urban or built-up area | 26421 | 0.47 |
| Agricultural or cultivated land | 69914 | 11.80 |
| Waterbody and marshland | 15815 | 2.85 |
| Forest or shrub | 16605 | 2.78 |
| Barren land and sand cover | 534504 | 81.29 |
| Permanent snow cover | 4167 | 0.76 |
| Total | 643646 | 100 |





Precipitation in the region is dominated by winter storms originating as eastward-moving Mediterranean cyclonic systems, which typically affect the country between November and April, peaking from January to March (Shokory et al., 2023). During summer, monsoonal airflows associated with the Intertropical Convergence Zone (ITCZ) cross the Afghanistan-Pakistan

border occasionally, bringing summer precipitation to the highest mountain peaks in the northeast. The highlands receive approx. 1000 mm/year between November and April (Aich et al., 2017; Shokory et al., 2023). By contrast, the lowland regions in the west and northern part of the country receive less than 150 mm/year (Aich et al., 2017; Shokory et al., 2023).

Climate observations indicate a strong trend of increasing mean annual temperature, which has increased by 1.8°C since the 1950s (NEPA, 2016). Projections suggest a likely increase by 1.4°C by 2030 (Shokory et al., 2023). Conversely, precipitation

has decreased by 1.5 to 6 mm/year in the spring season in southwestern and northeastern Afghanistan, whereas central, eastern, and southern Afghanistan has experienced slight increases in summer precipitation and in frequency of heavy (10 mm) and very heavy (20 mm) rainfall events (Aliyar et al., 2022; Shokory et al., 2023). This trend of increasing extreme precipitation events is likely to exacerbate the chances of flash flooding and associated landslides and mudslides, as well as f glacier lake outburst floods (GLOFs) (Bronkhorst and Bhandari, 2021; Sediqi et al., 2022; Shroder et al., 2022; Shroder and Weihs, 2010).

Decades of conflict and an unstable governance system have weakened societal resilience by altering the capacity of public institutions to combat natural disasters (UNCT, 2023). This combination of climatic and sociopolitical circumstances increases the vulnerability of Afghan communities to flood risks; economic losses due to flood hazards alone are estimated at $400 million annually (UNDP, 2023)

## 3    Data and Methods

Flood risk is defined here, as in previous studies, as the expected losses or harmful consequences resulting from the interactions between physical (hazard) events and anthropogenic (vulnerability) indicators (Alexander, 2000; Blaikie et al., 2014; Shahid and Behrawan, 2008). Our conceptual approach to flood risk assessment in Afghanistan therefore involves the combination of hazard and vulnerability analysis, with a separate assessment of their distinct influencing factors. Based on data availability at provincial level, a regional flood risk assessment for the whole of Afghanistan was performed. The general process of flood

risk assessment, as applied in this study, begins with the collection and analysis of primary and secondary data sets (Table 2).

**Table 2.** Details of primary and secondary datasets used for the analysis

| Data Type | Spatial resolution | Data Source |
|---|---|---|
| Digital elevation model | 90 m | http://hydro.iis.u-tokyo.ac.jp/~yamadai/MERIT_Hydro/ |
| Province level shapefiles | Scale = 1: 300,000,00 | https://www.igismap.com/download-afghanistan-administrative-boundary-gis-data-for-national-provinces-districts-and-more/ |




| District level shapefiles | Scale = 1: 300,000,00 | https://www.igismap.com/download-afghanistan-administrative-boundary-gis-data-for-national-provinces-districts-and-more/ |
|---|---|---|
| Census data Statistics | Province level | http://www.nsia.gov.af https://www.emro.who.int/child-adolescent-health/data-statistics/afghanistan.html |
| Precipitation datasets | 0.25° | https://opendata.dwd.de/climate_environment/GPCC/html/fulldata-monthly_v2022_doi_download.html |
| Landuse landcover data | Scale = 1: 50,000 | https://rds.icimod.org/Home/DataDetail?metadataId=1973187 |

These include the MERIT Hydro Digital Elevation Model (DEM), precipitation records, and socio-economic indicators such as census data and LULC data. We then further analysed these datasets by evaluating two main components: hazard and vulnerability indicators (Fig. 2). To enhance the understanding of the statistical calculation/analysing process flow, the driving 150 indicators of flood risk criteria i.e., hazard and vulnerability, are designated as Level-1 decision indicators. The sub-indicators within each of these indicators are classified as Level-2 decision indicators.

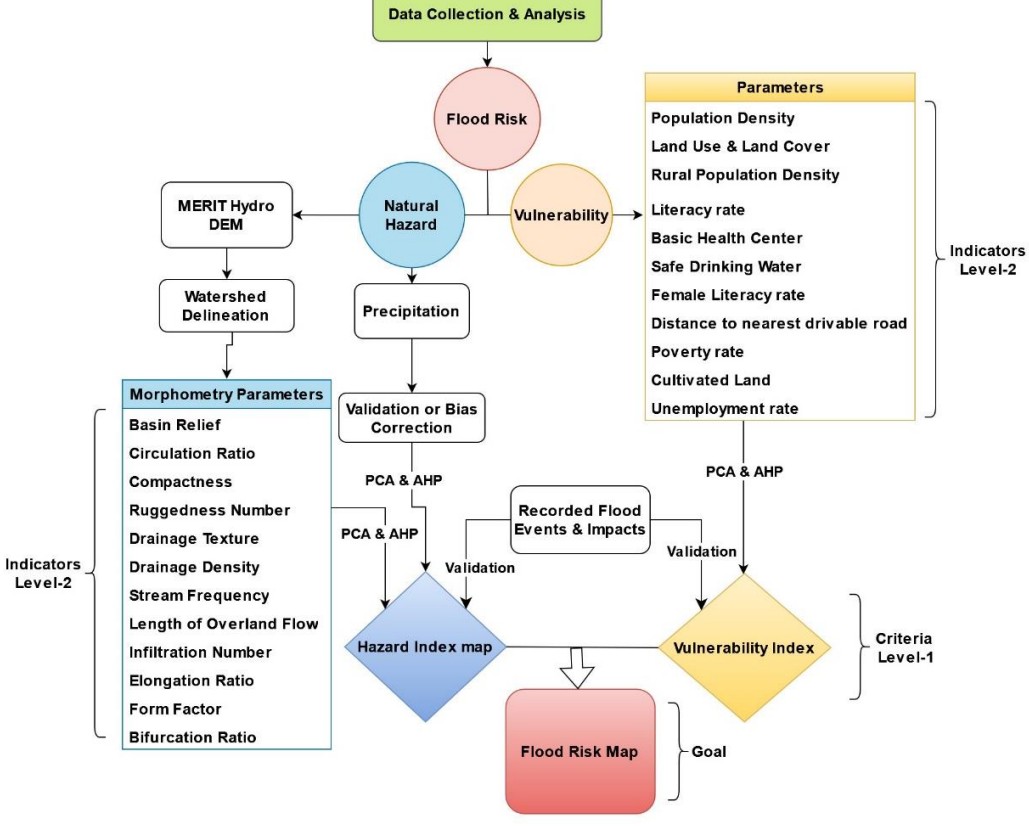





**Figure 2.** Shows the detailed flowchart used for the data collection, adopted methodology and analysis to generate the final hazard, vulnerability and risk index map

### 3.1 Datasets used

### 3.1.1 Flood hazard indicators

We used freely available 90 m MERIT Hydro DEM dataset obtained from the MERIT Hydro: global hydrography datasets (http://hydro.iis.u-tokyo.ac.jp/~yamadai/MERIT_Hydro/) to generate morphometric indicators by undertaking watershed delineation (Bharath et al., 2021; Koutroulis and Tsanis, 2010; Nasir et al., 2020; Samela et al., 2018) in Arc-GIS 10.8 tools. We identified 49 river basins across the country. For each of these, we computed 12 Level-1 morphometric indicators, namely drainage density, basin relief, drainage texture, infiltration number, stream frequency, compactness, circulation ratio, elongation ratio, bifurcation number, overland flow length, and form factor. This was undertaken using standard methods (see Supplementary Section S1.1 and formulae in Table 3). Each of these Level-1 indicators were then further classified into sub indicators (Level-2), as described in Tables 4a, S1, and S2.

**Table 3.** Formula and calculation of morphometric indicators for flood hazard index

| S.No. | Basin Parameters | Symbol (unit) | Formula | Reference |
|---|---|---|---|---|
| 1. | Area | $A$ (km$^2$) | $A$ = area of basin | |
| 2. | Length | $L_b$ (km) | $L_b$ = Length of basin | |
| 3. | Perimeter | $P_b$ (km) | $P_b$ = perimeter of basin | |
| 4. | Streams order | $N_0$ | Hierarchical rank | (Strahler, 1957) |
| 5. | Streams number | $N_u$ | $N_u = N_1 + N_2 + N_3 \ldots N_n$, where $N_u$ is number of streams of any given order | (Horton, 1945) |
| 6. | Streams length | $L_u$ (km) | $L_u = L_1 + L_2 \ldots L_n$ | (Strahler, 1957) |
| 7. | Stream frequency | $F_s$ | $F_s = N_u / A$ | (Horton, 1945) |
| 8. | Drainage density | $D_d$ (km/km$^2$) | $D_d = L_u / A$ | (Horton, 1945) |
| 9. | Basin relief | $B_r$ | $R = h - h_1$, where h = maximum height (km), $h_1$ = minimum height (km) | (Schumm, 1956) |
| 10 | Drainage texture | $D_t$ | $D_t = N_u / P$ | (Horton, 1945) |
| 11. | Infiltration number | $I_n$ | $I_n = F_s * D_d$ | (Faniran, 1968) |





| 12. | Compactness | $C_c$ | $C_c = 0.282 * P / \sqrt{A}$ | (Horton, 1945) |
|---|---|---|---|---|
| 13. | Circulation ratio | $C_r$ | $C_r = 4 * \pi * A / P^2$ | (Miller, 1953) |
| 14. | Elongation ratio | $E_r$ | $E_r = 2 * \sqrt{(A / \pi)} / L_b$ | (Schumm, 1956) |
| 15. | Ruggedness number | $R_n$ | $R_n = R * D_d$ | (Strahler, 1957) |
| 16. | Bifurcation | $B_r$ | $B_r = N_u / N_{u+1}$ | (Horton, 1945) |
| 17. | Length of overland flow | $LO_f$ | $L_o = 1 / (2 * D_d)$ | (Horton, 1945) |
| 18. | Form Factor | $F_f$ | $F_f = A / L_b^2$ | (Horton, 1945) |

Precipitation data were obtained from the Global Precipitation Climatology Centre at the Deutscher Wetterdienst (GPCC, http://gpcc.dwd.de/) (Table 2). We used mean monthly gridded data with a spatial resolution of 0.25° for the period 1990-2020 for the analysis. These outputs were further validated and bias-corrected using observational data provided by the Ministry of Water and Energy of Afghanistan. The precipitation data (Table 4a & Table S3) were then integrated with other indicators to assess the impact of rainfall patterns on potential flood hazard.

**3.1.2    Vulnerability indicators**

We obtained data related to socio-economic parameters from the Afghan National Statistical and Information Authority (NSIA; 2018 data as most recent available, http://www.nsia.gov.af), World Bank reports, and from the International Organization for Migration, (IOM) (https://www.emro.who.int/child-adolescent-health/data-statistics/afghanistan.html). These data sources were used to extract vulnerability indicators (Table 2). We note that due to the limited reliability of local documented data,
and the lack of demographic and census data at district or smaller scales, all indicators were obtained and quantified at the provincial level. This limitation challenges our ability to extract an inclusive vulnerability index from socio-economic parameters at finer scales. After careful consideration of the available and quantifiable parameters, we identified 10 Level-1 indicators available to us to derive a proxy vulnerability index: i.e., population density, rural population density, percentage of cultivated land, distance to nearest drivable road, female literacy rate, overall literacy rate, basic health centres, percentage of
households with access to safe drinking water, poverty rate, and unemployment (Fig. 2). Each of these Level-1 vulnerability indicators were further classified into sub indicators (Level-2) and are described in Tables 5a, S4, S5 and S6.

The Landuse and landcover (LULC) data were assessed using the National Land Cover Monitoring System (NLCMS) for Afghanistan, (https://rds.icimod.org/Home/DataDetail?metadataId=1973187). We reclassified these data into six major classes: (a) Urban or built-up areas; (b) Cultivated land; (c) Water bodies and marshlands (d) Barren and sand-covered land;
(e) Forests and shrubland; and (f) Snow-covered areas. These data were analysed to support the vulnerability index. The



percentages of the different reclassified LULC categories in Afghanistan are presented in Table 1. Vulnerability indicators are shown in Fig.2.

### 3.2    Statistical data analysis

The natural (physical) and socio-economic indicators of flood risk were analysed using Arc-GIS and remote sensing techniques. The output layer of corresponding indicators (Level-1) and sub indicators (Level-2) were converted into equal pixel-sized raster datasets to map spatial variability of hazard, vulnerability, and finally flood risk index. These indicators and sub indicators were then conditioned and rated using inputs from literature and decision-makers, including area experts, academia, and local governments (Table S7) (Danumah et al., 2016; Saaty, 1980, 1988; Stefanidis and Stathis, 2013). The initial conditioning was validated and corrected for bias using PCA (Ajtai et al., 2023; Maćkiewicz and Ratajczak, 1993).

Subsequently, the AHP (Saaty, 1980) was applied to refine and enhance the accuracy of the flood hazard and vulnerability maps. Finally, we used the natural breaks (Jenks) method (Jenks, 1967) to classify our maps into four major zonal classes: Low, Moderate, High and Very High zones.

**Table 4a.** Calculation of relative importance weightage (RIW) for decision indicators (Level – 1) for Flood hazard index (FHI)

| Decision Indicators | P | $B_R$ | $C_r$ | $C_c$ | $R_n$ | $D_t$ | $D_d$ | $F_s$ | $LO_f$ | $I_n$ | $E_r$ | $F_f$ | $B_r$ | EEV | RIW |
|---|---|---|---|---|---|---|---|---|---|---|---|---|---|---|---|
| Precipitation (P) | 1 | 2 | 3 | 3 | 3 | 3 | 4 | 4 | 4 | 4 | 5 | 5 | 6 | 3.33 | 0.20 |
| Basin Relief ($B_R$) | 1/2 | 1 | 2 | 2 | 2 | 2 | 3 | 3 | 3 | 3 | 4 | 4 | 5 | 2.30 | 0.14 |
| Circulation ratio ($C_r$) | 1/3 | 1/2 | 1 | 2 | 2 | 2 | 3 | 3 | 3 | 3 | 4 | 4 | 5 | 2.01 | 0.12 |
| Compactness ($C_c$) | 1/3 | 1/2 | 1/2 | 1 | 2 | 2 | 3 | 3 | 3 | 3 | 4 | 4 | 5 | 1.81 | 0.11 |
| Ruggedness number ($R_n$) | 1/3 | 1/2 | 1/2 | 1/2 | 1 | 2 | 3 | 3 | 3 | 3 | 4 | 4 | 5 | 1.62 | 0.10 |
| Drainage Texture ($D_t$) | 1/3 | 1/2 | 1/2 | 1/2 | 1/2 | 1 | 3 | 3 | 3 | 3 | 4 | 4 | 5 | 1.46 | 0.09 |
| Drainage Density ($D_d$) | 1/4 | 1/3 | 1/3 | 1/3 | 1/3 | 1/3 | 1 | 2 | 2 | 2 | 3 | 3 | 4 | 0.91 | 0.05 |
| Stream Frequency ($F_s$) | 1/4 | 1/3 | 1/3 | 1/3 | 1/3 | 1/3 | 1/2 | 1 | 2 | 2 | 3 | 3 | 4 | 0.82 | 0.05 |
| Length of overland flow ($LO_f$) | 1/4 | 1/3 | 1/3 | 1/3 | 1/3 | 1/3 | 1/2 | 1/2 | 1 | 2 | 3 | 3 | 4 | 0.74 | 0.04 |





| | | | | | | | | | | | | | | | |
|---|---|---|---|---|---|---|---|---|---|---|---|---|---|---|---|
| Infiltration number ($I_n$) | 1/4 | 1/3 | 1/3 | 1/3 | 1/3 | 1/3 | 1/2 | 1/2 | 1/2 | 1 | 3 | 3 | 4 | 0.66 | 0.04 |
| Elongation ratio ($E_r$) | 1/5 | 1/4 | 1/4 | 1/4 | 1/4 | 1/4 | 1/3 | 1/3 | 1/3 | 1/3 | 1 | 2 | 3 | 0.42 | 0.03 |
| Form Factor ($F_f$) | 1/5 | 1/4 | 1/4 | 1/4 | 1/4 | 1/4 | 1/3 | 1/3 | 1/3 | 1/3 | 1/2 | 1 | 2 | 0.37 | 0.02 |
| Bifurcation ($B_r$) | 1/6 | 1/5 | 1/5 | 1/5 | 1/5 | 1/5 | 1/4 | 1/4 | 1/4 | 1/4 | 1/3 | 1/2 | 1 | 0.7 | 0.02 |
| Sum of column | 4.40 | 7.03 | 9.53 | 11.03 | 12.53 | 14.03 | 22.42 | 23.92 | 25.42 | 26.92 | 38.83 | 40.50 | 53.00 | 16.72 | 1.00 |

**Table 4b.** Calculation of Consistency Ratio (CR) for decision indicators (Level – 1) of Flood hazard index (FHI)

| Decision Indicators | P | $B_R$ | $C_r$ | $C_c$ | $R_n$ | $D_t$ | $D_d$ | $F_s$ | $LO_f$ | $I_n$ | $E_r$ | $F_f$ | $B_r$ | Sum of rows | [E] |
|---|---|---|---|---|---|---|---|---|---|---|---|---|---|---|---|
| P | 0.23 | 0.28 | 0.31 | 0.27 | 0.24 | 0.21 | 0.18 | 0.17 | 0.16 | 0.15 | 0.13 | 0.12 | 0.11 | 2.57 | 12.89 |
| $B_R$ | 0.11 | 0.14 | 0.21 | 0.18 | 0.16 | 0.14 | 0.13 | 0.13 | 0.12 | 0.11 | 0.10 | 0.10 | 0.09 | 1.73 | 12.58 |
| $C_r$ | 0.08 | 0.07 | 0.10 | 0.18 | 0.16 | 0.14 | 0.13 | 0.13 | 0.12 | 0.11 | 0.10 | 0.10 | 0.09 | 1.52 | 12.65 |
| $C_c$ | 0.08 | 0.07 | 0.05 | 0.09 | 0.16 | 0.14 | 0.13 | 0.13 | 0.12 | 0.11 | 0.10 | 0.10 | 0.09 | 1.38 | 12.75 |
| $R_n$ | 0.08 | 0.07 | 0.05 | 0.05 | 0.08 | 0.14 | 0.13 | 0.13 | 0.12 | 0.11 | 0.10 | 0.10 | 0.09 | 1.25 | 12.90 |
| $D_t$ | 0.08 | 0.07 | 0.05 | 0.05 | 0.04 | 0.07 | 0.13 | 0.13 | 0.12 | 0.11 | 0.10 | 0.10 | 0.09 | 1.14 | 13.08 |
| $D_d$ | 0.06 | 0.05 | 0.03 | 0.03 | 0.03 | 0.02 | 0.04 | 0.08 | 0.08 | 0.07 | 0.08 | 0.07 | 0.08 | 0.73 | 13.36 |
| $F_s$ | 0.06 | 0.05 | 0.03 | 0.03 | 0.03 | 0.02 | 0.02 | 0.04 | 0.08 | 0.07 | 0.08 | 0.07 | 0.08 | 0.66 | 13.55 |
| $LO_f$ | 0.06 | 0.05 | 0.03 | 0.03 | 0.03 | 0.02 | 0.02 | 0.02 | 0.04 | 0.07 | 0.08 | 0.07 | 0.08 | 0.60 | 13.71 |
| $I_n$ | 0.06 | 0.05 | 0.03 | 0.03 | 0.03 | 0.02 | 0.02 | 0.02 | 0.02 | 0.04 | 0.08 | 0.07 | 0.08 | 0.55 | 13.82 |
| $E_r$ | 0.05 | 0.04 | 0.03 | 0.02 | 0.02 | 0.02 | 0.01 | 0.01 | 0.01 | 0.01 | 0.03 | 0.05 | 0.06 | 0.35 | 13.93 |
| $F_f$ | 0.05 | 0.04 | 0.03 | 0.02 | 0.02 | 0.02 | 0.01 | 0.01 | 0.01 | 0.01 | 0.01 | 0.02 | 0.04 | 0.30 | 13.44 |
| $B_r$ | 0.04 | 0.03 | 0.02 | 0.02 | 0.02 | 0.01 | 0.01 | 0.01 | 0.01 | 0.01 | 0.01 | 0.01 | 0.02 | 0.22 | 13.54 |
| Sum of column | 1.00 | 1.00 | 1.00 | 1.00 | 1.00 | 1.00 | 1.00 | 1.00 | 1.00 | 1.00 | 1.00 | 1.00 | 1.00 | 13.00 | 172.20 |

$\lambda_{max} = 13.25$; CI= 0.02; RI= 1.56; CR=0.01





**Table 5a.** Calculation of relative importance weightage (RIW) for decision indicators (Level – 1) of flood vulnerability index (FVI)

| Decision Indicators | $P_d$ | LULC | $P_{rd}$ | $L_r$ | $H_c$ | $DW_p$ | $FL_r$ | $DD_r$ | $C_L$ | $P_r$ | $U_r$ | EEV | RIW |
|---|---|---|---|---|---|---|---|---|---|---|---|---|---|
| Population Density ($P_d$) | 1 | 2 | 2 | 3 | 3 | 3 | 3 | 4 | 4 | 5 | 7 | 3.01 | 0.21 |
| LULC | 1/2 | 1 | 2 | 2 | 3 | 3 | 3 | 4 | 4 | 5 | 6 | 2.52 | 0.18 |
| Rural Population Density ($P_{rd}$) | 1/2 | 1/2 | 1 | 2 | 3 | 3 | 3 | 4 | 4 | 4 | 5 | 2.14 | 0.15 |
| Literate rate ($L_r$) | 1/3 | 1/2 | 1/2 | 1 | 3 | 3 | 3 | 4 | 4 | 4 | 4 | 1.78 | 0.13 |
| Basic Health centers ($H_c$) | 1/3 | 1/3 | 1/3 | 1/3 | 1 | 2 | 2 | 3 | 3 | 3 | 3 | 1.13 | 0.08 |
| Safe Drinking Water ($DW_p$) | 1/3 | 1/3 | 1/3 | 1/3 | 1/2 | 1 | 2 | 2 | 2 | 3 | 3 | 0.93 | 0.07 |
| Female Literate ($FL_r$) | 1/3 | 1/3 | 1/3 | 1/3 | 1/2 | 1/2 | 1 | 2 | 2 | 2 | 3 | 0.79 | 0.06 |
| Distance to nearest drivable road ($DD_r$) | 1/4 | 1/4 | 1/4 | 1/4 | 1/3 | 1/2 | 1/2 | 1 | 2 | 2 | 2 | 0.58 | 0.04 |
| Cultivated Land ($C_L$) | 1/4 | 1/4 | 1/4 | 1/4 | 1/3 | 1/2 | 1/2 | 1/2 | 1 | 2 | 2 | 0.51 | 0.04 |
| Poverty Rate ($P_r$) | 1/5 | 1/5 | 1/4 | 1/4 | 1/3 | 1/3 | 1/2 | 1/2 | 1/2 | 1 | 2 | 0.42 | 0.03 |
| Unemployment Rate ($U_r$) | 1/7 | 1/6 | 1/5 | 1/4 | 1/3 | 1/3 | 1/3 | 1/2 | 1/2 | 1/2 | 1 | 0.33 | 0.02 |
| Sum of column | 4.18 | 5.87 | 7.45 | 10.00 | 15.33 | 17.17 | 18.83 | 25.50 | 27.00 | 31.50 | 38.00 | 14.15 | 1.00 |

**Table 5b.** Calculation of Consistency Ratio (CR) for decision indicators (Level – 1) of flood vulnerability index (FHI)

| Decision Indicators | $P_d$ | LULC | $P_{rd}$ | $L_r$ | $H_c$ | $DW_p$ | $FL_r$ | $DD_r$ | $C_L$ | $P_r$ | $U_r$ | Sum of rows | [E] |
|---|---|---|---|---|---|---|---|---|---|---|---|---|---|
| $P_d$ | 0.24 | 0.34 | 0.27 | 0.30 | 0.20 | 0.17 | 0.16 | 0.16 | 0.15 | 0.16 | 0.18 | 2.33 | 10.95 |
| LULC | 0.12 | 0.17 | 0.27 | 0.20 | 0.20 | 0.17 | 0.16 | 0.16 | 0.15 | 0.16 | 0.16 | 1.91 | 10.73 |
| $P_{rd}$ | 0.12 | 0.09 | 0.13 | 0.20 | 0.20 | 0.17 | 0.16 | 0.16 | 0.15 | 0.13 | 0.13 | 1.63 | 10.79 |
| $L_r$ | 0.08 | 0.09 | 0.07 | 0.10 | 0.20 | 0.17 | 0.16 | 0.16 | 0.15 | 0.13 | 0.11 | 1.40 | 11.11 |
| $H_c$ | 0.08 | 0.06 | 0.04 | 0.03 | 0.07 | 0.12 | 0.11 | 0.12 | 0.11 | 0.10 | 0.08 | 0.91 | 11.29 |
| $DW_p$ | 0.08 | 0.06 | 0.04 | 0.03 | 0.03 | 0.06 | 0.11 | 0.08 | 0.07 | 0.10 | 0.08 | 0.74 | 11.25 |
| $FL_r$ | 0.08 | 0.06 | 0.04 | 0.03 | 0.03 | 0.03 | 0.05 | 0.08 | 0.07 | 0.06 | 0.08 | 0.62 | 11.19 |
| $DD_r$ | 0.06 | 0.04 | 0.03 | 0.03 | 0.02 | 0.03 | 0.03 | 0.04 | 0.07 | 0.06 | 0.05 | 0.47 | 11.37 |
| $C_L$ | 0.06 | 0.04 | 0.03 | 0.03 | 0.02 | 0.03 | 0.03 | 0.02 | 0.04 | 0.06 | 0.05 | 0.41 | 11.34 |



| | | | | | | | | | | | | | |
|---|---|---|---|---|---|---|---|---|---|---|---|---|---|
| $P_r$ | 0.05 | 0.03 | 0.03 | 0.03 | 0.02 | 0.02 | 0.03 | 0.02 | 0.02 | 0.03 | 0.05 | 0.33 | 11.17 |
| $U_r$ | 0.03 | 0.03 | 0.03 | 0.03 | 0.02 | 0.02 | 0.02 | 0.02 | 0.02 | 0.02 | 0.03 | 0.25 | 10.79 |
| Sum of column | 1.00 | 1.00 | 1.00 | 1.00 | 1.00 | 1.00 | 1.00 | 1.00 | 1.00 | 1.00 | 1.00 | 11.00 | 121.97 |
| $\lambda_{max}$ = 11.09; CI= 0.01; RI= 1.51; CR=0.01 | | | | | | | | | | | | | |

### 3.2.1 Principle Component Analysis (PCA)

Principal Component Analysis (PCA) is a statistical data analysis technique used to reduce the number of variables in a dataset by transforming a large set of interrelated variables into a smaller set, while preserving as much of the original information as

possible (Ajtai et al., 2023; Nandi et al., 2016; Wu, 2021). In this study, PCA was performed using IBM SPSS statistical software: Level -1 indicators were fed into the software to calculate the loading impact of each indicator on the principal components of hazard and vulnerability by grouping highly correlated indicators. The number of principal components was determined based on eigenvalue, which is typically greater than 1. In order to increase the interpretability of the results, we applied Kaiser's Varimax rotation (European Union, 2008; Kaiser and Rice, 1974).

Based on the identified linear associations, we established the ranking order of indicator loadings on the corresponding hazard and vulnerability indices (Fig. 2; Table 6 & Table 7). This was followed by assigning relative importance weightage (RIW) using the AHP. The PCA approach was also employed to assess the key indicators influencing the hazard, vulnerability, and risk indices.

**Table 6.** PCA based ranking of flood hazard indicators using Varimax with Kaiser Normalization methods

| Flood hazard indicators | Component loading (%) | | | |
|---|---|---|---|---|
| | 1 (42 %) | 2 (20 %) | 3 (16 %) | 4 (7 %) |
| Precipitation | 0.823 | | | |
| Basin Relief | 0.782 | | | |
| Circulation ratio | 0.784 | | | |
| compactness | -0.801 | | | |
| Ruggedness number | 0.747 | | | |
| Drainage texture | 0.697 | | | |
| Drainage density | | 0.967 | | |
| Stream frequency | | 0.712 | | |
| Length of overland flow | | -0.879 | | |
| Infiltration number | | 0.881 | | |
| Elongation ratio | | | 0.951 | |





| Form factor | | | 0.872 | |
| Bifurcation | | | | 0.971 |


**Table 7.** PCA based ranking of flood vulnerability indicators using Varimax with Kaiser Normalization methods

| Flood vulnerability indicators | Component loading (%) | | | |
| --- | --- | --- | --- | --- |
| | 1 (33 %) | 2 (17 %) | 3 (13 %) | 4 (10 %) |
| Population Density | 0.798 | | | |
| LULC | 0.692 | | | |
| Rural population Density | 0.660 | | | |
| Literate rate | 0.636 | | | |
| Basic health center | 0.626 | | | |
| Safe drinking water | 0.606 | | | |
| Female literacy rate | | 0.732 | | |
| Distance to nearest drivable road | | 0.679 | | |
| Cultivated land | | | 0.603 | |
| Poverty rate | | | 0.517 | |
| Unemployment rate | | | | -0.702 |

### 3.2.2    Analytical hierarchy process (AHP)

The AHP (Saaty, 1980) is a widely used Multi-Criteria Decision-Making (MCDM) technique (Danumah et al., 2016; Shehata
and Mizunaga, 2018; Triantaphyllou, 2000; Vojtek and Vojteková, 2019). It incorporates both qualitative and quantitative
factors ensuring logical consistency and minimizing bias in prioritizing indicators (Ahmed et al., 2024; Danumah et al., 2016;
Vojtek and Vojteková, 2019). It provides a systematic approach for evaluating and integrating the impact of various indicators,
by defining the relative importance and weights of each indicator. The AHP process involves a) selecting the indicators, b)
constructing the hierarchy, c) creating the pairwise comparison matrix, d) assigning numerical values to compute weights, and
e) validating the consistency ratio.

The pairwise comparison matrix is the most crucial step in AHP, whereby each Level-1 and Level-2 criterion (Fig.2) is
compared against others using the Saaty numbers scale ranges from 1 (equal importance) to 9 (extreme importance) (Table
S7). The values assigned to the upper half of the matrix represent decision maker's subjectivity, experience, and knowledge
(Ikirri et al., 2022; Mokhtari et al., 2023; Saaty, 1980; Vojtek and Vojteková, 2019). The lower half of the matrix uses the
inverse of the values assigned to upper half (Chakraborty and Mukhopadhyay, 2019; Saaty, 1988).

After developing the pairwise matrix, we then calculated Estimated Eigen Values (EEV) in the matrix for each indicator and
sub indicators. This was achieved using Eq. (1).

$$EEV = \sqrt[N]{W_{1*}W_{2*}W_3 \dots W_N} \qquad (1)$$




Where $W_1$, $W_2$, $W_3$, $W_N$ are the weightage of the row elements in the matrix tables. These weightages are assigned to the different indicators during a pairwise comparison at Level-1 and Level-2 criterion to evaluate their relative importance (Tables 4a, 5a, Tables (S1- S6)). The prioritization of these factors at Level-1 and Level-2, is based include area experts, stakeholders from academia, local government representatives, and local community members (Mishra and Sinha, 2020; Radwan et al., 2019; Stefanidis and Stathis, 2013). N is the number of the row elements or the number of compared indicators. The relative ranking of Level-1 and Level-2 criterion was retained using PCA analysis which explained 85% and 73% of the total variance for hazard and vulnerability index respectively (Table 6 and Table 7).

The second step is to compute the Relative Importance Weights (RIW) of each criteria using Eq. (2). As a rule, the total sum of RIW in each matrix of indicators and sub indicators must be equal to one.

$$RIW = \frac{\sqrt[N]{W_1*W_2*W_3...W_N}}{EEV_1+EEV_2+EEV_3...EEV_N} \tag{2}$$

Where $EEV_1$, $EEV_2$, $EEV_3$, $EEV_N$ are the estimated eigen value of each element in an individual matrix.

The third step is to check consistency to avoid calculation errors and ensure a realistic matrix. This is obtained by determining the consistency ratio (CR) (Table 4b and Table 5b).

$$CR = \frac{CI}{RI} \tag{3}$$

Where CI is the Consistency Index and RI is the Random Index, a value that depends on the number of criteria, as recommended by (Saaty, 1988) (Table S8).

The Consistency Index (CI) is calculated using:

$$CI = \frac{\lambda_{max}-n}{n-1} \tag{4}$$

Where n is the number of indicators and $\lambda_{max}$ is the largest eigen value that is determined from normalization matrix ($\lambda_{max}$) by computing sum of the products between each element of the RIW and column totals of the matrix as Eq. (5).

$$\lambda_{max} = \frac{[E]}{n} \tag{5}$$

where [E] is the rational priority, defined as:

$$[E] = \frac{\sum of\ row\ of\ normalization\ matrix}{RIW} \tag{6}$$

A reasonable level of consistency is considered when the CR does not exceed 0.10, If the CR exceeds 0.10, indicating inconsistency, the pairwise comparison ratings should be adjusted to achieve the required consistency (Saaty, 1980).



The consistency ratio for both the Flood hazard index (FHI) and Flood vulnerability index (FVI) parameters were calculated. Once completed, the final maps were then generated through an arithmetic overlay of all criteria as expressed by Eq. (7) and (8) respectively.

$$\textbf{\textit{FHI}} \;=\; 0.2 * P + 0.14 * B_R + 0.12 * C_r + 0.11 * C_c + 0.10 * R_n + 0.09 * D_t + 0.05 * D_d + 0.05 * F_s + 0.04 * LO_f + 0.04 * I_n + 0.03 * E_r + 0.02 * F_f + 0.02 * B_r \tag{7}$$

$$\textbf{\textit{FVI}} = 0.21 * P_d + 0.18 * LULC + 0.15 * P_{rd} + 0.13 * L_r + 0.08 * H_c + 0.07 * DW_p + 0.06 * FL_r + 0.04 * DD_r + 0.04 * C_L + 0.03 * P_r + 0.02 * U_r \tag{8}$$

Finally, the Flood Risk Index (FRI) map was produced by multiplying FHI and FVI maps at province level as presented in Eq. (9).

$$Flood\ Risk\ Index\ (FRI) = Flood\ Hazard\ Index * Flood\ Vulnerability\ Index \tag{9}$$

The respective FHI, FVI and FRI maps identify regions at various hazard, vulnerability and risk levels. These maps show a continuous range of values, which were further classified into four major classes: Class-I (Very High), Class-II (High), Class-III (Moderate), Class-IV (Low), based on the Jenks natural break method. The idea is to classify the data intervals with a natural break method, which optimizes data grouping by minimizing intra-class variance while maximizing inter-class variance, thereby enhancing the distinction between different data clusters (Lin, 2013). We consider this as best approach for mapping spatial data, as it identifies natural patterns, enhances interpretability, and supports better decision-making in risk assessment.

### 3.2.3    Result comparison

The results of flood hazard and associated vulnerability for the current study were compared using a database documenting flood incidents and their socio-economic impacts for the period 2019-2024. Flood incident and impact data were gathered from local authorities, news outlets, and United Nations datasets, and provide information at province level. In excess of 600 flood events were considered. Their spatial frequency was visualized across the study area to validate our flood hazard results. We used socio-economic damages and losses from these events to verify our flood vulnerability findings. In present section, we assess whether the methodology and data analysis processes were consistent and accurate at provincial level. This evaluation aims to confirm the suitability of the method for further applications in natural and climatic studies, particularly in data-scarce regions.





# 4 Results

## 4.1 Flood Hazard Index (FHI)

We generated a continuous FHI map using 12 indicators (Level-1) and their respective sub-indicators (Level-2). FHI values range from 1.78 (low) to 4.15 (high) and are spatially variable (Fig.3a). Figure 3b illustrates the spatial variability of FHI
across the country using four major categories. The relative ranking and calculation process of the Level-1 and Level-2 decision indicators, based on PCA and AHP, are presented in Table 4a and 5a, Tables S1(a-f), Table S2(a-f), and S3. Details about indicator selection are further discussed in supplementary sections (S3).

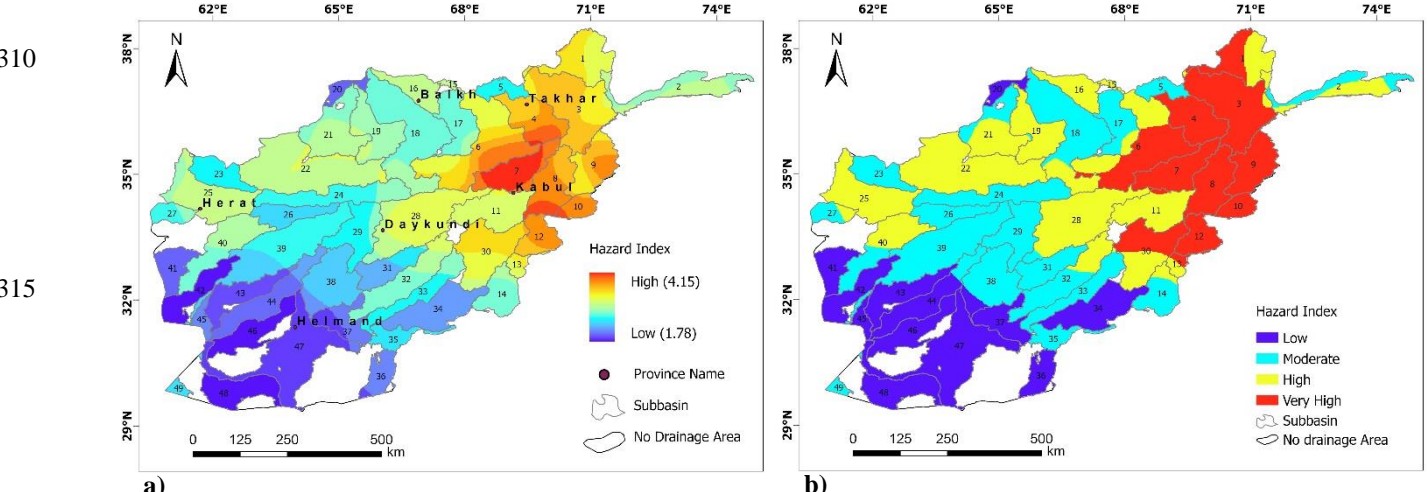



**Figure 3:** a) Flood Hazard Index (FHI) map;  b) Classified Flood Hazard Index (FHI) map

Approximately 20% (1.19 * $10^6$ sq.km) of the study area falls within the Very High flooding zone, while 29% (1.72 * 106 sq.km) is prone to regular flooding. In contrast, areas associated with low to moderate flooding susceptibility are distributed over 20% (1.19 * 106 sq.km) and 31% (1.84 * $10^4$ sq.km) of the total area, respectively. These classes were further used to
identify the susceptibility of each basin to different FHI zones/ classes.

Some catchments contain multiple sub-basins which may each yield different hazard classes. Depending on their spatial distribution, some catchments yield a single hazard class, while others share two different hazard classes. For example, subbasin numbers 3, 4, 7, 8, 9, 10, and 12 are associated with the Amu and Kabul River basins, and are assessed as having Very High flood risk. Within those same catchments, subbasins 1, 6, 13, and 30 yield both Very High and High FHI. Subbasin
numbers 15, 16, 19, 22, 25, and 28, which are part of the Harirud-Morghab and Northern River basins, are exposed to High flood risk. Some subbasins, such as 14, 18, 21, 27, 32, and 40, also lie across two flood zones within their respective regions. The least hazardous area lies in the southwestern Helmand basin (including subbasin numbers 24, 26, 31, 35, 37, 38, 39, 41, 42, 43, 47), which is assessed to have Moderate to Low flood risk (Fig.3b).





To identify the major controlling factors associated with the different hazard zones, all morphometric and hydrological
indicators were evaluated using PCA at the subbasin level. PCA identifies the following class characteristics:

**Class-I:** "Very High flooding zone" areas are linked to basin relief, short overland flow lengths, high infiltration number, and dense drainage networks, by cumulative loading of 37%. A high circulation ratio and low compactness ratio indicate more circular basins, which can influence the concentration time and peak discharge during floods with a 19% loading. Precipitation in areas with specific basin shapes, such as high form factor and elongation ratio, results in a quicker runoff response and
potential for flash floods, influencing flood hazard by 18% of loadings.

**Class-II:** "High flooding zones" are controlled by the shape and compactness of the basin. High circulation ratio, form factor, and elongation ratio, combined with low compactness, suggest that this component captures the geometrical characteristics of the basin and influences a High flooding hazard zone by 32%. The hydrological properties of the basin, such as high drainage density, infiltration number, and low basin relief and length of overland flow, indicate regions with dense drainage networks
and significant infiltration number, influencing High flood hazard by 31%. Precipitation and stream bifurcation values contribute 14%, suggesting that areas with frequent branching of streams and high precipitation are more prone to flooding.

**Class -III:** "Moderate flooding zones" are dominated by basin shape and relief associate with 31% impact load. High elongation ratio and form factor values indicate a more circular basin shape, while basin relief suggests significant vertical relief that lead to rapid surface runoff. The drainage characteristics of the basins, such as high drainage density, shorter length
of overland flow, higher infiltration number and rugged terrain account for 28% load within Moderate flooding zones. High precipitation and low compactness contribute an additional 16%.

**Class-IV:** "Low flooding zones" are influenced by drainage characteristics and overland flow, influencing 39% of impact load. High drainage texture, stream frequency, drainage density and infiltration number indicate dense networks of drainage channels with reduced overland flow, suggesting shorter flow paths with quick surface runoff. The high elongation ratio and
form factor which represent the shape of the basin, captures 27% of the flood loading, causing a quicker runoff response and potential for flash floods. The basin compactness and relief contribute 23% to low flooding zone loading.

#### 4.1.1 Comparison of flood hazard maps

To validate the flood hazard findings, the documented flood incident reports from the United Nations datasets (https://data.humdata.org/dataset) between 2019 and 2024 were used. Approximately 613 recorded flood incidents across the
country at the province level were visualized (Fig.4). The provinces with the highest flood events (39-70) are concentrated in northeastern and eastern Afghanistan, including Badakhshan, Takhar, Kunar, and Nangarhar, which correspond to the areas marked as extremely high flood hazards on the first map. Moderate flood incident zones (12-23 incidents, orange) are distributed across central, northern, and western Afghanistan, while provinces in the southwest, such as Kandahar, Helmand, and Nimroz, experience fewer flood incidents (yellow), aligning with their classification as low hazard regions in the first map.
Comparing the classified hazard map with recorded flood incidents, the eastern (Kabul River basin) and northeastern (Amu River basin) parts of the study area experienced the highest number of flooding events that includes subbasin number 1, 3, 4,



8, 9, and 10. Similarly, subbasin number 20, 21, 22, 23, 25, 27, and 40 lies in the northern (Northern River basin) and northwestern (Harirud-Morphab River basin) regions were identified as the second most flooded zones, while the subbasin number 28, 39, 41, 42, 43, and 44 associated with central and southwestern (Hilmand River basin) regions were categorized as rarely flooded zones. Such comparison of the flood hazard map with flood incidents suggests the consistency of the flood hazard approach model used in the study.

## 4.2   Flood Vulnerability Index (FVI)

The continuous vulnerability scale of FVI with a range of low as 3.32 to high as 9.91 was derived using eleven indicators namely, population density, rural population density, percentage of cultivated land, distance to nearest drivable road, female literacy rate, general literacy rate, basic health centres, percentage of household safe drinking water, poverty rate, unemployment rate and LULC illustrated in Fig.5a. The final FVI map, shown in Fig.5b, was classified using the same methodological approach as FHI at level-1 and in level-2 (Table 5a, Table S4 and S5) into 4 major classes.  Similar to FHI map, the FVI map, RIWs were computed, and the CR was found to be satisfactory (i.e. C.R ≤ 0.10 or ≤ 10 %) as shown in Table 5b.

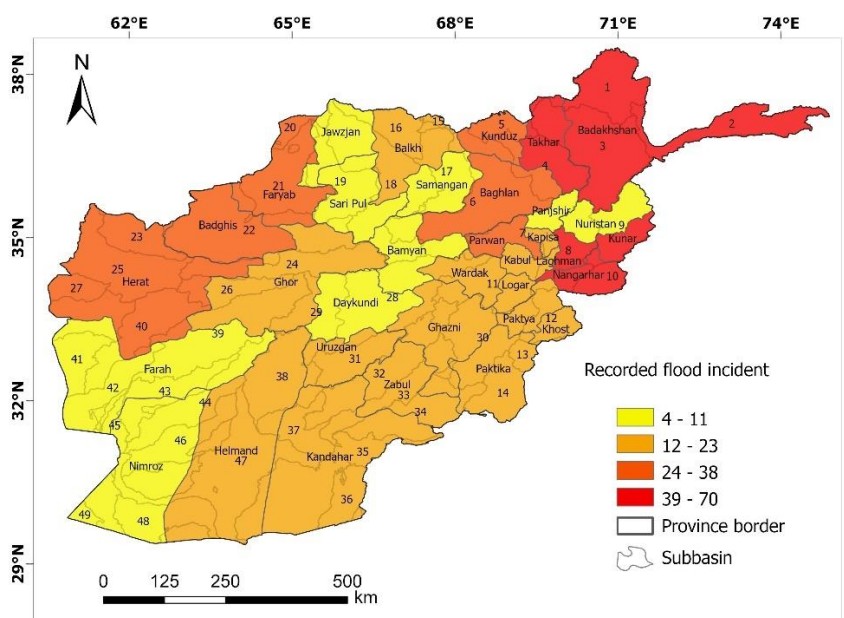

**Figure 4.** Recorded flood events in the Afghanistan region at province level

The results indicate that approximately 31% (1.84 * $10^6$ sq.km) of the study are, located primarily in the northern and northwestern regions of Afghanistan, falls within Very High vulnerability zone. The north-eastern and central regions are categorized as Highly vulnerable zone covering 20% (1.19 * $10^6$ sq.km) of the study area. In contrast, the southern and western



plain regions are classified into moderate and low vulnerable zone covering 31% (1.84 * $10^6$ sq.km) and 17% (1.01 * $10^6$
sq.km) of the area respectively. Based on the spatial distribution of the subbasin number at Province level, some fall into a
single FVI class, while others share two different FVI classes within the study area.

To identify the major controlling factors associated with the different vulnerability zones, all the eleven indicators were
evaluated using PCA approach in subbasin levels. The PCA results indicates that:

**Class-I:** "Very High vulnerability zone" are characterised with high population density, high poverty rates, and poor road
accessibility are both highly exposed and vulnerable to flooding, contributing to 32% of the Very High vulnerability zones.
Significant land use changes and intensive agricultural activity account for 24% of the Very High vulnerability areas. Changes
in land cover exacerbate flood risks by increasing runoff and reducing natural water absorption within the soil.

**Class-II:** "High vulnerability zone" is primarily influenced by socioeconomic factors and accessibility. Areas with high
population densities, better access to health centres and safe drinking water, lower female literacy rates, and greater distances
from drivable roads are the main controlling indicators, contributing 48% to the loading impact. Additionally, areas with lower
literacy rates and higher amounts of cultivated land account for 22% of the loading impact, affecting their resilience to floods.

**Class-III:** "Moderate vulnerability zone" is characterized by high population densities, both urban and rural, better
accessibility to safe drinking water, and locations farther from drivable roads, which increase vulnerability due to reduced
accessibility for evacuation and emergency services, contributing 33% to the loading impact. Factors such as education, land
use changes, and unemployment are also significant, accounting for 26% of the loading impact on the Moderate vulnerability
zone.

**Class-IV:** "Low vulnerability zone" represented by high population density and land use changes that control over 57% of

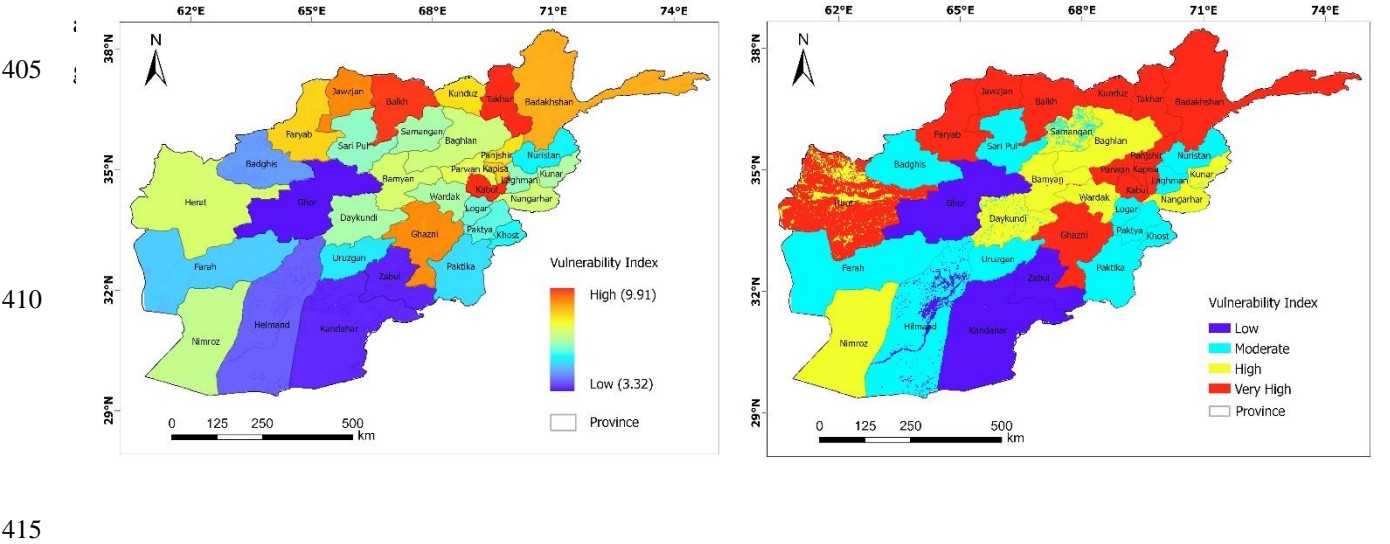

a)                                                                                          b)

**Figure 5 : a) Flood Vulnerability Index (FVI) map; b) Classified Flood Vulnerability Index (FVI) map**



#### 4.2.1 Comparison of Flood Vulnerability Maps

The flood vulnerability index results were validated by using same reported flood incident consequences datasets
(https://data.humdata.org/dataset ) as used in flood hazard index. According to flood events statistics of loss of life and injuries
during the documented period, more than 2000 people has lost their life or were seriously injured. Majority of fatalities were
recorded in Kunar, Nuristan, Nangarhar, Parwan, Baghlan, Ghor, Frayab and Herat provinces in eastern, north-eastern and
north-western part of the study area. However, the northern provinces of Balk, Jawzjan, Samangan and Bamyan and southern
and south-western provinces of Paktika, Paktia, Nimruz and Farah is recorded with few fatalities (Fig. 6).

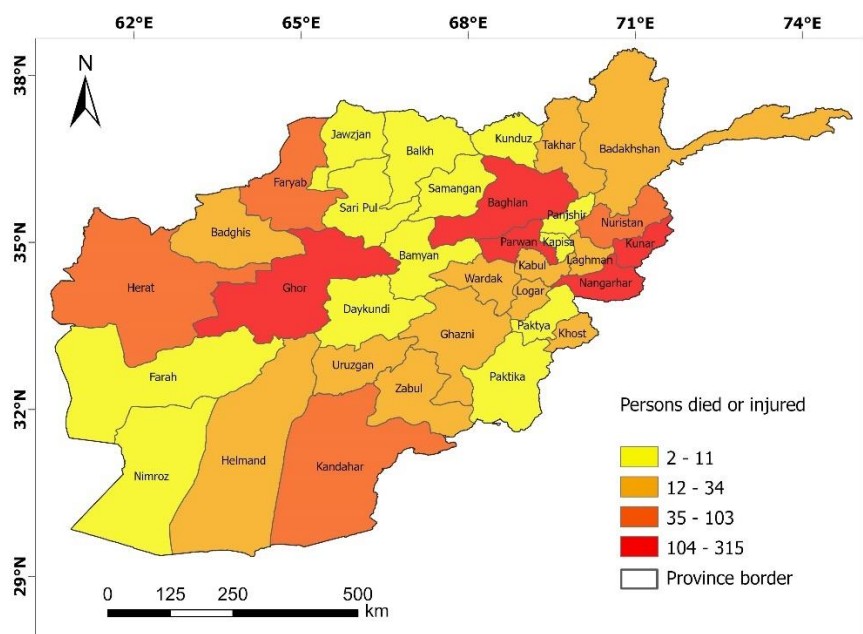


**Figure 6.** Total number death and injuries during various flooding events (between 2019 and 2024)

### 4.3 Flood Risk Index (FRI)

The floods risk map of the Afghanistan region was prepared by combining the flood hazard and flood vulnerability maps as
shown using Eq. (9). Both FHI and FVI maps were multiplied in the ArcMap model builder to generate the flood risk map
(Fig.7a), that was further classified into four risk zones/class using natural Jenks classification method (Fig. 7b and 7c). The
final classified FRI map indicates that 24% ($1.42 * 10^6$ sq.km) of the study area, falls within Very High flood risk zone. The
High risk zone covering 22% ($1.31 * 10^6$ sq.km) of the study area. In contrast, Moderate and Low vulnerable zone covering
26% ($1.54 * 10^6$ sq.km) and 28 % ($1.66 * 10^6$ sq.km) of the area respectively. Based on the spatial distribution at Basin/subbasin
scale and Province/district scale, some of the region fall into a single FRI class, while others share two different FRI classes
within the study area.



Based on classified FRI map (Fig.7b) at subbasin level, subbasin numbers 1, 3, 4, 6 within Amu River Basin, major parts of subbasin number 15, 16, 18 (Northern basin) 7 and 10 (Kabul basin) are associated with Very High flood risk zone. Similarly, subbasins, such as 23, 25, 26 and 27 (Harirud Morghab basin), 28, 40 (Helmand basin), 8, 9, 11, 12 (Kabul basin) are exposed to High FVI zones. In contrast, the Northern and Helmand basin, including subbasin numbers 13, 14, 20, 22, 31, 32, 44, is

dominated by Moderate risk zone followed by subbasin number 24, 26, 33, 34, 35, 38, 42, 47 in Low flood risk zones. Sub basins such as 2, 6, 8, 9, 17,18, 21 of different basins span two flood vulnerability zones within their respective regions.

Similarly, FRI map at Province level (Fig.7c) suggests that the northeastern province such as Badakshan, Takhar, Panjher, major parts of Baglan, Bailh, Kabul lies in Very High flood risk zone, whereas Herat in west, major parts of Dayakundi and Bamiyan in middle, Khost, Paktik, Vardhak, Kunuar province in the east are associated with high FRI zone. Similarly, south

and southwestern part of the Afghanistan region is majorly dominated by moderate and low FRI zone. Such examples include Zabol, Quandhar, Helman, Nimbruz, Frahan, Ghowr province.

To identify the major controlling factors associated with the different risk zones at subbasin and province level, all the associated thirteen hazard and eleven vulnerability indicators were evaluated using PCA approach. The PCA results summarised in Table S9, and it shows that:

**Class-I:** "Very High flood risk Zone" indicates the subbasins within high stream frequency, circular basins, higher precipitation, low compact basin and higher infiltration number values governs the Very-High risk. These areas are physically exposed to floods due to increased runoff and rapid water accumulation. In terms of socioeconomic factors, regions with high population density, significant land use changes, better access to safe drinking water, and high poverty rates play a significant role in determining Very High risk zones (Table S9).

**Class-II:** "High flood risk zone", is dominated with overland flow length and drainage characteristics of the subbasins. Areas with longer overland flow paths and significant basin relief within areas with lower drainage density and infiltration numbers governs High risk flooding zone. Higher population density, including rural populations, better access to safe drinking water, and lower literacy rates (both general and female) are key socioeconomic factors contributing to High flooding zone. This suggests that higher population densities and better access to drinking water are more vulnerable, while low literacy rates

exacerbate this vulnerability, highlighting the need for educational improvements to reduce flood risk (Table S9).

**Class-III:** "Moderate flood risk zone" is characterized by higher drainage density, shorter overland flow, and higher infiltration numbers, indicating low water absorption capacity. Socioeconomic parameters include higher rural and overall population density, proximity to drivable roads, significant land use changes, and lower female literacy rates. This suggests that densely populated areas even with better road access, but lower female literacy rates may be more vulnerable to floods due to high

population pressure and land use changes (Table S9).

**Class-IV:** "Low flood risk zone" is dominated with higher drainage density, lower overland flow and higher infiltration number that suggest characteristics like higher surface runoff potential or less effective infiltration, which could contribute to increased flood risk. Literate rate, closer to accessible road and lower population density reduced flood risk in Low-risk zone areas (Table S9).






**Figure 7:** a) Flood Risk Index (FRI) map; b) Classified Flood Risk Index (FRI) map at subbasin level; and c) Zonation of
classified FRI map at provincial level

## 5    Discussions

### 5.1    Factors affecting flood risk assessment and future directions

The results of present study highlight the complex interplay between geomorphological, hydrological, and socio-economic
factors in determining flood risk across Afghanistan region. By integrating the FHI and FVI, we enable to establish a
comprehensive FRI map that captures the spatial variability of flood risk at subbasin and province level.

Based on the present FHI analysis approximately 20% of the study area falls within a "Very High" flood hazard zone, primarily
concentrated in the northeastern and eastern regions of Afghanistan, such as the Amu and Kabul River basins. These regions
are characterised by high basin relief, dense drainage networks, and short overland flow lengths, which contribute to rapid
water accumulation and increased runoff during heavy precipitation events. Conversely, the southwestern Helmand basin
exhibited predominantly "Moderate" to "Low" flood risk. This region's relatively flat topography, coupled with lower
precipitation levels and higher infiltration rates, contributes to reduced flood hazard levels. The spatial variability in flood



hazard is consistent with findings from other semi-arid regions where topography and precipitation play a critical role in determining flood risk patterns (Nabinejad and Schuttrumpf, 2023; Sanyal and Lu, 2004). The study also highlights the influence of morphometric characteristics on flash flood intensity (Nazeer and Bork, 2019; Sediqi and Komori, 2023; Taha et al., 2017; Waqas et al., 2021). They also play crucial role in predicting the response of each watershed during periods of heavy rainfall (Abdel-Fattah et al., 2017; Lombana et al., 2024; Mahmood and Rahman, 2019) emphasizing the importance of terrain characteristics in flood risk assessments (Borga et al., 2011). The comparison with the present flood incident maps obtained from various sources comprehends the reliability of morphometric parameters together with other satellite parameters, acting as proxy to replicate the flood hazard maps in the data limited region (Manawi et al., 2020; Sediqi et al., 2022; Shroder and Ahmadzai, 2016).

Similarly, the FVI results indicate that socio-economic factors significantly contribute to flood vulnerability across Afghanistan. Areas with high population density, low literacy rates, poor road accessibility, and high poverty levels are more vulnerable, as observed in the northeastern provinces of Baghlan, Takhar, and Badakhshan, which fall under the "Very High" vulnerability zone, which clearly aligned with the previous studies (Omerkhil et al., 2020; Trani et al., 2010). This aligns with findings from global flood risk studies, which emphasize that socio-economic disparities often exacerbate the impacts of natural disasters (Bouaakkaz et al., 2023; Cutter, 1996; El Kharraz et al., 2012). Moreover, changes in land use, particularly increased agricultural activities and urban expansion, have reduced natural water absorption capacity, leading to higher runoff during flood events. Similar patterns have been reported in studies from regions undergoing rapid land-use transformation, where increased impervious surfaces lead to heightened flood risks (Zope et al., 2017).

The integration of hazard and vulnerability indices to create the FRI map provides a more holistic view of flood risk across the region. The FRI results indicate that 24% of Afghanistan, particularly the northeastern and eastern parts, is classified as a "Very High" risk zone. Sub-basins within the Amu and Kabul River basins, such as those identified by sub-basin numbers 1, 3, and 6, are notably susceptible to severe flood events. The high risk in these areas is primarily driven by a combination of geomorphological factors and socio-economic vulnerabilities, including high population density and poor infrastructure (Klijn et al., 2015; Radwan et al., 2019). In contrast, provinces such as Helmand, Nimroz, and Farah in the southern and southwestern parts of Afghanistan show a lower flood risk, attributed to flatter terrain, lower population density, and better accessibility to essential services. These findings echo previous research, which has highlighted the role of socio-economic resilience, including infrastructure and education, in mitigating flood risk (Erena and Worku, 2018; Zhou and Liu, 2024).Limitations and future directions

The approach employed in this study is particularly useful for regions where long-term hydrological data is scarce enabling flood risk assessments using alternative indicators such as morphometric parameters and socio-economic factors (Biswas, 2004; Mishra and Sinha, 2020; Teng et al., 2017). However, several difficulties were encountered, which highlights the limitations of consistent, high-quality scientific literature and well-documented observational data, which complicates the validation of results and comparison across different regions (Hirpa et al., 2013; Zhou and Liu, 2024). A major challenge is the lack of field measurements and data availability in arid and semi-arid zones of developing countries, where catchments are



often poorly gauged and hydro-meteorological data is insufficient (Ahmed et al., 2023; Nabinejad and Schuttrumpf, 2023). Additionally, deficiencies in census data at smaller administrative levels and difficulties in integrating multiple datasets with varying formats and spatial resolutions pose challenges for developing comprehensive models (Albano et al., 2020).

The absence of consistent observed hydrological data further limits the ability to directly compare different satellite datasets, impacting the ability to select appropriate data sources for hydrological analysis. This challenge affects the verification of precipitation's role in flood hazards, as demonstrated in studies by Mazzoleni et al., (2018) and Shukla et al., (2020), where accurate rainfall data was crucial for model calibration. For regions with complex topography, such as Afghanistan, which displays significant elevation differences from a few meters to thousands of meters, high-quality data processing and robust analysis techniques are essential to accurately delineate morphometric parameters (Nabinejad and Schuttrumpf, 2023; Rahmati

et al., 2019). The diversity in topography complicates the understanding of flood dynamics, necessitating localized analysis to capture the variability in flood responses effectively.

Moreover, the response of morphometric parameters to flood risk varies significantly across different sub-basins, being influenced by local hydro-morphometric characteristics, census data, land cover, and soil structures (Bharath et al., 2021; Taha et al., 2017; Youssef et al., 2016). For example, in regions such as Badakhshan province in the northeast, multiple sub-

subbasins exhibit varying hydro-morphometric conditions, meaning that identical census data values can lead to different flood risk responses depending on local topographical and hydrological characteristics (Hosseini et al., 2022). This variability highlights the importance of conducting flood risk assessments at both subbasin and provincial scales, integrating local factors such as soil characteristics, land cover, and drainage patterns (Vojtek and Vojteková, 2019).

Future studies should prioritize refining data collection, expanding observational networks, and developing methodologies

adaptable to data-scarce environments. Improved integration of remote sensing data with localized field observations can help overcome data gaps, while advances in GIS processing capabilities can enhance the accuracy of morphometric parameter extraction in regions with diverse topography (Rahmati et al., 2019). Additionally, sensitivity analysis is necessary to understand how specific indicators and weighting factors influence the final classification, which can mitigate uncertainties inherent in the current approach (Pamukçu Albers and Evers, 2024). Ultimately, these efforts will improve the robustness and

transferability of the methodology to other data-scarce regions, enabling more reliable flood risk assessments globally.

### 5.2 Implications on flood risk management

The findings of this study have significant implications for flood management strategies in Afghanistan. The identified "Very High" and "High" risk zones should be prioritized for flood mitigation efforts, such as constructing flood barriers, improving drainage systems, and establishing early warning systems. Additionally, enhancing socio-economic resilience through public

awareness programs, literacy initiatives, infrastructure improvements, and increased access to healthcare can play a crucial role in reducing vulnerability.

Considering local resources, traditional methods, climatic characteristics, and governing indicators of the subbasins, the recommended measures for Very High and High flood zones within the provinces of Badakhshan, Takhar, and Baghlan (Amu



River basin) include terracing, floodplain restoration, check dams, and flood storage reservoirs. Implementing these structural measures in these provinces, where flood drivers include steep basin relief, short, intense precipitation, and rapid, unregulated urbanization, is expected to reduce flood risk and provide safer areas for residents.

In the Panjshir, Kapisa, Parwan, Kabul, Nuristan, Nangarhar, and Kunar provinces (Kabul River basin), which are characterized by basin circularity, rapid runoff, sudden and irregular precipitation, and extensive land cover changes, terracing, flood reservoirs, and soil conservation are highly recommended. Additionally, afforestation and increased vegetation cover are encouraged in both the Amu and Kabul river basins to improve flood management and reduce sediment transport, as the local climatic conditions are favourable for these practices. Channel improvement and rainwater harvesting are recommended for the Balkh, Herat, Faryab, Ghazni, and Wardak provinces, where the low basin slope, high drainage density, and low overland flow dominate the region.

Protecting indigenous plants and wild trees is strongly recommended across the country to recharge groundwater, reduce erosion, and decrease flood intensity. Similar approaches have been successfully implemented in other disaster-prone regions, where comprehensive disaster risk reduction strategies have significantly reduced flood impacts (UNISDR, 2015). Non-structural and organizational flood protection measures are also recommended across Afghanistan, including strengthening community resilience, enhancing public awareness, adopting sustainable land-use practices, and establishing a systematic national flood action plan.

However, flood management policies face specific challenges in arid and semi-arid regions like Afghanistan, which differ significantly from those in humid areas due to the unique characteristics of floods and the technical and operational precautions required. These challenges are further intensified by limited organizational resources, low levels of risk awareness, and limited community engagement (Nabinejad and Schuttrumpf, 2023).

The complex climatic and morphometric conditions of the study area including spatial and temporal variability in precipitation, prolonged droughts, and sparse vegetation strongly impact the types of flood risk strategies that can be effectively implemented. For example, extensive erosion and sediment transport, driven by loose soils and inadequate vegetation cover, often cause serious damage to flood protection structures, reducing their functionality. Moreover, the lack of reliable data, weak institutional support, limited public awareness, and low community engagement present significant obstacles to executing non-structural measures.

Given these conditions, the flood hazard and risk maps produced in this study serve as essential tools to guide authorities in implementing sustainable flood management strategies that are carefully adapted to the conditions at both the subbasin and provincial levels.

## 6 Conclusions

This study presents a comprehensive flood risk assessment for Afghanistan as a whole, using an integrated hydro-morphological approach that combines RS and GIS technologies. Unlike traditional hydrological models, which primarily rely



traditional hydrological or hydraulic models, this method integrates physical, social, economic, and anthropogenic factors based on open-source remote sensing datasets to assess flood hazards and vulnerabilities. We consider this approach to offer an effective adaptable solution for large-scale areas with limited hydrological data. It also enables replication across regions, making it a valuable tool for flood risk management in data-scarce environments.

We find that the Amu and Kabul River basins in eastern and northeastern Afghanistan are exposed to very high flood hazards. This is due to the combined effects of precipitation, topography, and drainage characteristics, which contribute to rapid runoff and increased flooding potential, especially during the Indian summer monsoon season. We categorise the central, northern, and northwestern regions as having high to moderate flood hazard. This result is due to the high degree of well-developed drainage networks – although flooding risks can be increased by the presence of short overland flow paths and low infiltration

rates. We observe lower flood hazards in the western and southwestern regions due to reduced precipitation, gentler slopes, and higher bifurcation ratios that disperse floodwaters, mitigating flood risks.

The combination of physical flood hazard assessment with vulnerability highlights the importance of socioeconomic factors in flood risk in a country like Afghanistan. Our vulnerability assessment indicates that high population density, especially in rural areas, along with significant land use changes, produce the greatest susceptibility to flood risk. The northern and eastern

regions tick all these boxes and are the most susceptible to flood risk. By contrast, in the south and southwest, there is a lower population density and predominantly barren lands, resulting in reduced flood vulnerability. These insights are crucial for government and non-governmental organizations to prioritize resource allocation and target interventions to mitigate potential impacts by better understanding the spatial distribution of flood risks. We therefore recommend an integrated approach to identify key factors influencing flood hazards and vulnerabilities. Although we maintain that this approach is the best possible

given the current data availability for this region, we emphasise that data availability and quality present significant challenges. Afghanistan represents perhaps one of the most extreme examples, as a region with particularly complex topography combined with political instability. Model accuracy can be enhanced and uncertainties reduced by improving data collection, infrastructure, and monitoring networks. The sensitivity of our methodology to input data and weighting factors highlights the need for robust validation and sensitivity analyses to ensure reliable outcomes.

Future efforts should focus on developing sustainable flood management strategies that incorporate both hazard and vulnerability assessments. Enhancing integrated flood management approaches and investing in resilient infrastructure can reduce flood risks and build resilience in vulnerable communities. Overall, this study provides a valuable framework for assessing flood risks, which can be adapted and applied in other regions facing similar environmental and data-related challenges, encouraging more effective disaster management and mitigation practice.

**7    Data availability**

Aggregated data can be acquired from the first author.



## 8 Appendix and Supplementary data

## 9 Author contributions

QI and KM conceptualized the research and determined the methodology. QI led the investigation and wrote the original draft
of the paper. KM, CZ and KF supported the interpretation and documentation of the results.

## 10 Competing interests

The authors declare that they have no conflict of interest.

## 11 Acknowledgements

We gratefully acknowledge the financial support provided by the University of Central Asia (UCA), the German Academic
Exchange Service (DAAD), the Federal Ministry of Education and Research (BMBF), and the Baden-Württemberg Ministry
of Science as part of the Excellence Strategy of the German Federal and State Governments, awarded to KM (Grant PRO-
MISHRA-2023-12) for funding this study. Additionally, we appreciate the support of the Open Access Publishing Fund of the
University of Tübingen.

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
