# Peer review of "Flood hazard in Afghanistan is intensified both by natural and socioeconomic factors"

_EGUsphere, 2025_

## Referee Comment (RC1)

**Review Comments**

1. The title is indicating the outcome of this study, it must be revised.

2. The authors sometimes have indicated the objective as identifying flood risk, sometimes as hazard and sometimes as the vulnerability. This is confusing. It is suggested to fix the objective first.

3. In the background, need clarification of the significance of this study and highlight research gap that should be addressed.

4. Concise the Title of Figure 2.

5. Why have you used rural population density along with population density?

6. You have shown Flood Hazard Index (FHI) map in Figure 3(a) however, justify the purpose and significance of using Classified Flood Hazard Index (FHI) map as shown in Figure 3(b).

7. Requires similar justification for Figure 4 and 5.

8. What do mean by the effects of topography, and drainage characteristics?

9. In the result section you have indicated 'Our vulnerability assessment indicates that high population density, especially in rural areas, along with significant land use changes, produce the greatest susceptibility to flood risk'

10. In general, the urban area faces frequent landuse change and occupy high population, however your finds differ from this one. Can you please justify.

---

## Referee Comment (RC2)

**Review report**

**Flood hazard in Afghanistan is intensified both by natural and socioeconomic factors**

**General comments:**

The paper evaluates flood hazard in Afghanistan by combining natural and socioeconomic indicators with remote sensing, as well as GIS-based, methods. The authors seek to identify flood-risk areas in Afghanistan, with an emphasis upon the contributing physical and social factors

One of the salient points in favor of the research is contextual appropriateness; Afghanistan ranks among the globe's most climate-exposure countries, and research in terms of spatial flood hazard in such a context is timely. The reliance on openly available datasets as well as GIS software provides a replicable process for other low-data countries. Yet, there exist a variety of shortcomings that have to be rectified. First, novelty is poor. The use of multi-criterial assessment of flood hazard based on topographic, hydrological, as well as socioeconomic indicators is an ordinary process in literature. The authors fail to explicitly show in what ways their research improves past methods or presents a novel process specific to Afghanistan. Second, the methodology was not made transparent. The selection process of indicators, weights, as well as their integration into hazard as well as vulnerability indices, is unclearly described. Significant information on data normalization, reclassification, or thresholding is missing or underexplained. Third, validation of the outputs is poor. The flood hazard as well as vulnerability maps are shown in a manner that lacks sufficient quantitative validation, including comparison against historic flood occurrence records or ground-truth datasets. Finally, the figures as well as maps have poor cartographic quality as well as insufficient integration into text, missing precise legends or explanation. These shortcomings in total necessitate a complete overhaul of the manuscript.

**Specific Comments by section:**

**Abstract:**

The abstract provides a broad description of the paper but does not describe the methods employed in detail. The use of terms like "this study assesses flood risk" and "critical insights for policymakers" does not enlighten the reader about which tools, datasets, or analysis have been used. The contribution of remote sensing, Principal Component Analysis (PCA), and Analytical Hierarchy Process (AHP) must explicitly be stated. The conclusion section of the abstract can more effectively point towards key findings along with quantitative information instead of broader points. The abstract, in total, needs to be rewritten for specificity, clarity, and conciseness.

**Introduction:**

The introduction gives a sound contextual description of global and regional flood threat trends, supplemented by a number of appropriate citations. The rationale for conducting this research, however, remains tenuously developed. Although Afghan vulnerability is strongly highlighted, the literature gap remains unexplicit. Assertions of sparse prior research in Afghanistan lack foundation in a systematic review of the literature. The claim to newness is also exaggerated, considering that

comparable flood hazard mapping with GIS and socioeconomic factors is widely established in the literature. The closing paragraphs could profit by having explicit articulation of objectives, research questions, and anticipated outcomes, which in current text lack or exist only obliquely.

**Study Area:**

The study area description section is good and organized, giving a description of Afghanistan's topography, climate, river basins, and land use. The authors succeed in describing the heterogeneity in terms of both physicality and climate in the country. The labeling in Figure 1 should, however, be made clear, as should corresponding references in writing. The information in Table 1 is informative but should have better formatting, as well as identification of sources. Better discussion of ways in which these factors impact flood risk spatially would make such a section more pertinent to the research's objectives.

**Data and Methods:**

This section is by far the weakest in the paper and needs to be rewritten. Though a broad set of indicators is set out for both vulnerability and hazard assessment, it is not adequately explained why they have been chosen. The description of data sources, especially remote sensing datasets, should be more specific—sensor type, resolution, temporal span, and pre-processing steps should be outlined. The PCA and AHP are cited as tools to be used for analysis, but which exact steps to perform these analyses are opaque. It is not possible to determine from which indicators have been standardized, which thresholds have been used, or which validation methods have been employed to rank indicators. The PCA description omits detail in terms of component extraction, variance captured, or interpreting factor loadings. The PCA section mentions using IBM SPSS and eigenvalue criteria but lacks details on the number of indicators, how they were selected, and the thresholds for retaining components.The process of constructing AHP matrices is in table format, but description in text format lacks detail, and it becomes an abstract process. The application of AHP states that weights were derived but does not specify how pairwise comparisons were made (e.g., participants, criteria, and consistency ratio).The process of integrating PCA and AHP results into flood hazard maps needs clearer explanation, including criteria for classification thresholds. Additionally, though a calculation of the consistency ratio (CR) occurs, its significance goes unexplained. The final equation of the risk index is given with no rationale as to why weights have been selected or why the hazard map should be multiplied by vulnerability map. Figure 2 illustrating a clear methodological flowchart exists, but each step in said diagram needs to be described in text. Without transparency, the ability to replicate the research is diminished.

**Results:**

Results are presented descriptively, mostly as a narrative to supplement the maps. The PCA-based ranking of indicators (Table 6) is valuable, but the explanation of what the component loadings represent, their physical interpretation, and thresholds for significance are insufficient. The flood hazard and vulnerability maps (Figures 3-5) are critical outputs. However, their validation is either missing or insufficiently described. The risk zones are identified with natural break classification, which is appropriate, but the criteria for defining "High" and "Very High" zones should be clarified. Findings of spatial distributions for different subbasins and provinces are reported, but little explanation exists for why these areas have a higher level of risk beyond cursory understanding. Also, the provided maps have not been designed to publication standards—legends are obscure, font

sizes differ, and color choices are hard to distinguish. The authors reference validation from 600 records of flood events, but the comparison still exists only visually, in a qualitative way. Quantitative validation measures such as accuracy, sensitivity, or confusion matrix calculations would add a great deal of strength to findings. There exists no statistical correlation or regression analysis to examine relationships among individual hazard/vulnerability indicators and flood occurrence observations.

**Discussion:**

There is little critical examination of the findings. The authors reiterate largely descriptive findings instead of interpreting them more substantively or comparing them with other regional/world studies. The claim that "this study provides a valuable framework" should be tempered with discussion of its limitations, especially regarding validation and uncertainty. The implications for policy or flood management are briefly mentioned but need elaboration.The impact of land cover transformation, infrastructure growth, and institutional capacity towards flood vulnerability is cited, though these factors haven't been substantiated with concrete data or strict analysis. The limitations of the data (e.g., time resolution, aged census data) and methodology (e.g., assumptions in PCA or AHP) aren't addressed. The discussion also fails to include policy-relevant observations specific to Afghanistan's institutional or geographical context.

**Conclusion:**

The conclusion section is generic and lacks analytical depth. Statements are broad and not directly tied to the study's specific findings. The authors should summarize the main results using quantitative evidence and outline specific recommendations based on the risk map. The conclusion should also acknowledge the limitations of the study and suggest directions for future research.

---

## Author Comment (AC2)

**Detailed Response to the Referee # 2 comments**

**General Comments:**

The paper evaluates flood hazard in Afghanistan by combining natural and socioeconomic indicators with remote sensing, as well as GIS-based, methods. The authors seek to identify flood-risk areas in Afghanistan, with an emphasis upon the contributing physical and social factors.

One of the salient points in favor of the research is contextual appropriateness; Afghanistan ranks among the globe's most climate-exposure countries, and research in terms of spatial flood hazard in such a context is timely. The reliance on openly available datasets as well as GIS software provides a replicable process for other low-data countries. Yet, there exist a variety of shortcomings that have to be rectified. First, novelty is poor. The use of multi-criterial assessment of flood hazard based on topographic, hydrological, as well as socioeconomic indicators is an ordinary process in literature.

The authors fail to explicitly show in what ways their research improves past methods or presents a novel process specific to Afghanistan. Second, the methodology was not made transparent. The selection process of indicators, weights, as well as their integration into hazard as well as vulnerability indices, is unclearly described. Significant information on data normalization, reclassification, or thresholding is missing or underexplained. Third, validation of the outputs is poor. The flood hazard as well as vulnerability maps are shown in a manner that lacks sufficient quantitative validation, including comparison against historic flood occurrence records or ground-truth datasets. Finally, the figures as well as maps have poor cartographic quality as well as insufficient integration into text, missing precise legends or explanation. These shortcomings in total necessitate a complete overhaul of the manuscript.

**Response:**

Thank you very much for your insightful and constructive comments. We appreciate your recognition of the importance of this study in a climate vulnerable and data scarce country such as Afghanistan. While multi criteria flood risk assessment is a well-established approach in the literature, to the best of our knowledge, this is the first time such an integrated method combining hazard and vulnerability indices using openly available datasets has been applied in Afghanistan. Given the absence of spatial flood risk assessments in this context, we believe this study provides a novel and practical framework that can guide future hydroclimate research in the country. It can help the public better understand flood prone zones and raise awareness of associated risks in

their daily lives. It can also support new researchers by providing opportunities to gain practical skills using the tools and methods applied in this study.

The methodology has been explained as clearly and concisely as possible, considering the publication's page limitations. As a general rule, risk is understood as the combination of physical hazards and socioeconomic vulnerability, as described in the manuscript. There are various other literatures that have used this approach to calculate risk. Some of the references are as follows:

1. Stefanidis, S., & Stathis, D. (2013). Assessment of flood hazard based on natural and anthropogenic factors using analytic hierarchy process (AHP). Natural hazards, 68, 569-585.

2. Mokhtari, E., Mezali, F., Abdelkebir, B., & Engel, B. (2023). Flood risk assessment using analytical hierarchy process: A case study from the Cheliff-Ghrib watershed, Algeria. Journal of Water and Climate Change, 14(3), 694-711

3. Radwan, F., Alazba, A. A., & Mossad, A. (2019). Flood risk assessment and mapping using AHP in arid and semiarid regions. Acta Geophysica, 67, 215-229.

4. Vojtek, M., & Vojteková, J. (2019). Flood susceptibility mapping on a national scale in Slovakia using the analytical hierarchy process. Water, 11(2), 364.

Since the study area is located in a region with limited data availability, we relied on hazard and vulnerability indicators derived from remote sensing data products, along with existing datasets referenced in the literature and available reports for the region. We have carefully outlined the key steps involved in ranking these indicators using Principal Component Analysis (PCA), followed by classification and weighting through the Analytical Hierarchy Process (AHP), both well-established methods in multi-criteria decision-making. These methodological details have been concisely presented to remain within the article's page limit while ensuring clarity and reproducibility.

We have also improved our explanation of data normalization, reclassification, and thresholding to enhance transparency. Afghanistan is a vast region with diverse geographic setting, that limits the feasibility of ground truth validation, especially given its current global and regional challenges. To address concerns regarding validation, we have incorporated comparisons between our hazard and vulnerability outputs and historical flood occurrence records, as well as reported property losses. These comparisons are based on available government statistics, official records, and published literature in this data-limited region. Such comparisons at the regional and national scales provide a reasonable level of confidence in our results. In addition, high-quality and cartographically enhanced figures will be included in the final version of the manuscript. We are confident that these clarifications and revisions comprehensively address the concerns raised

and significantly enhance the contribution of our study, particularly given the scarcity of literature and ground data for a country like Afghanistan.

**Specific Comments by section:**

**Abstract:**

**Comments:** The abstract provides a broad description of the paper but does not describe the methods employed in detail. The use of terms like "this study assesses flood risk" and "critical insights for policymakers" does not enlighten the reader about which tools, datasets, or analysis have been used. The contribution of remote sensing, Principal Component Analysis (PCA), and Analytical Hierarchy Process (AHP) must explicitly be stated. The conclusion section of the abstract can more effectively point towards key findings along with quantitative information instead of broader points. The abstract, in total, needs to be rewritten for specificity, clarity, and conciseness.

**Response:** Thank you very much for your feedback. In response to your comments and considering the word limit constraints for the abstract, we have revised it as follows: Taking above mentioned comments into account, we have revised the abstract as follows:

*"The increasing frequency of climate driven extreme events such as heavy precipitation, floods, and droughts poses severe social, economic, and environmental challenges. Among these, floods are particularly destructive, causing substantial damage to lives, property, and infrastructure. This study assesses flood risk in Afghanistan by integrating remote sensing data and geographic information systems to evaluate flood hazard and vulnerability at both sub-basin and provincial levels. Principal Component Analysis is used to identify the most influential environmental, climatic, and socioeconomic indicators, while the Analytical Hierarchy Process is applied to weigh these indicators in a structured and consistent manner. The results indicate that the eastern and northeastern regions of Afghanistan especially within the Amu and Kabul River basins are most prone to flood hazards due to intense precipitation, steep topography, and rapid surface runoff. Vulnerability analysis reveals that densely populated rural areas in the north and east are particularly at risk, largely due to land use changes and limited adaptive capacity. These results provide essential guidance for local communities and national authorities by identifying high flood risk zones and supporting focused measures to enhance flood preparedness and resilience".*

We would also like to mention that, due to word limitations in the abstract, it is not feasible to provide detailed methodological explanations. However, we have made every effort to concisely include the key tools and findings as per your suggestion.

**Introduction:**

**Comments:** The introduction gives a sound contextual description of global and regional flood threat trends, supplemented by a number of appropriate citations. The rationale for conducting this research, however, remains tenuously developed. Although Afghan vulnerability is strongly highlighted, the literature gap remains unexplicit. Assertions of sparse prior research in Afghanistan lack foundation in a systematic review of the literature. The claim to newness is also exaggerated, considering that comparable flood hazard mapping with GIS and socioeconomic factors is widely established in the literature. The closing paragraphs could profit by having explicit articulation of objectives, research questions, and anticipated outcomes, which in current text lack or exist only obliquely.

**Response:** Agreed. In response to your comments, we have revised lines 65-83 to better address the literature gap, justify the chosen methodology, and clearly outline the research objectives as follows:

*"Although Afghanistan is highly vulnerable to flooding, a comprehensive national level flood risk mapping that integrates both environmental and socioeconomic indicators has not yet been performed. Moreover, there is a notable lack of literature addressing flood risk assessment at this scale within the country. This gap underscores the critical need for a systematic assessment of flood sensitivity across Afghanistan.*

*Regular monitoring of a large, poorly resourced country such as Afghanistan is not only time consuming but also logistically difficult and adds to its financial burden (Goyal et al., 2020). For data-scarce regions, remote sensing (RS) and spatial data analysis with Geographic Information Systems (GIS) has emerged as an effective tool (Bhatt et al., 2014). A range of satellite derivatives and GIS techniques, many of which are open-source, are now available to assess entire country profiles for flood hazard and sensitivity (Holand et al., 2011; Membele et al., 2022; Mokhtari et al., 2023; Ogarekpe et al., 2020; Samela et al., 2018). This includes the recent development of remotely sensed hydro-morphological characterization over large spatial scales which overcomes the requirement for long-term hydrological data to incorporate into traditional hydrological models (Teng et al., 2017).*

*Therefore, the overall objective of this study is to map and assess flood risk in Afghanistan region by integrating hazard and vulnerability components at subbasin level. Given the mountainous nature of the country and its administrative division into broadly hydro-morphically defined provinces, we approach this by 1) Assessing flood hazard and vulnerability indices for each hydro-morphologically defined sub-basin using associated environmental, topographic, and socio-economic indicators; 2) Identifying and ranking the key environmental and climatic parameters*

*that drive the flood risk using principal component analysis (PCA) and 3) Defining the relative importance of these parameters through the Analytical Hierarchy Process (AHP) to assign weights and establish risk levels. The integrated approach will highlight the regions which are most susceptible to flooding and to understand the underlying causes of their vulnerability. The resultant maps are intended to serve as a basis for government and relief agencies to develop appropriate management plans. By incorporating social vulnerability into assessments for flood management, decision-makers can more effectively allocate resources to those communities most in need".*

**Study Area**

**Comments:** The study area description section is good and organized, giving a description of Afghanistan's topography, climate, river basins, and land use. The authors succeed in describing the heterogeneity in terms of both physicality and climate in the country. The labeling in Figure 1 should, however, be made clear, as should corresponding references in writing. The information in Table 1 is informative but should have better formatting, as well as identification of sources. Better discussion of ways in which these factors impact flood risk spatially would make such a section more pertinent to the research's objectives.

**Response:** We appreciate your positive feedback on the study area description. Following your suggestions, we have updated the labeling and improved the clarity of Figure 1; the revised figure will be included in the final manuscript. All figures were generated using primary and secondary datasets, which are detailed in Table 2 (line 146). Table 1 has been reformatted and revised, with its areal statistics derived from Land Use and Land Cover (LULC) data, also referenced in Table 2 (line 146). Additionally, a more detailed discussion on how these factors spatially influence flood risk has been added in Supplementary Section S2, subsection 2.11 (Land Use and Land Cover).

**Data and Methods**

**Comments:** This section is by far the weakest in the paper and needs to be rewritten. Though a broad set of indicators is set out for both vulnerability and hazard assessment, it is not adequately explained why they have been chosen. The description of data sources, especially remote sensing datasets, should be more specific—sensor type, resolution, temporal span, and pre-processing steps should be outlined. The PCA and AHP are cited as tools to be used for analysis, but which exact steps to perform these analyses are opaque. It is not possible to determine from which indicators have been standardized, which thresholds have been used, or which validation methods have been employed to rank indicators. The PCA description omits detail in terms of

component extraction, variance captured, or interpreting factor loadings. The PCA section mentions using IBM SPSS and eigenvalue criteria but lacks details on the number of indicators, how they were selected, and the thresholds for retaining components. The process of constructing AHP matrices is in table format, but description in text format lacks detail, and it becomes an abstract process. The application of AHP states that weights were derived but does not specify how pairwise comparisons were made (e.g., participants, criteria, and consistency ratio). The process of integrating PCA and AHP results into flood hazard maps needs clearer explanation, including criteria for classification thresholds. Additionally, though a calculation of the consistency ratio (CR) occurs, its significance goes unexplained. The final equation of the risk index is given with no rationale as to why weights have been selected or why the hazard map should be multiplied by vulnerability map. Figure 2 illustrating a clear methodological flowchart exists, but each step in said diagram needs to be described in text. Without transparency, the ability to replicate the research is diminished.

**Response:** We thank you for your suggestions. The hazard and vulnerability indicators, along with their pre-processing steps, are more thoroughly explained in Supplementary Sections 1 and 2. For detailed descriptions of the data sources, including remote sensing datasets, please refer to Table 2 (line 146).

PCA and AHP were employed as tools to minimize subjectivity and enhance consistency through the grouping, ranking, and weighting of indicators, which are explained in Sections 3.2.1 and 3.2.2 of the manuscript.

In general, the selection of hazard and vulnerability indicators in this type of study is based on the unique environmental, topographical, socio-economic characteristics, and data availability of the study area. The ranking and weighting of each indicator were performed using relevant literature as well as input from local decision-makers, domain experts, and academics, as outlined and revised in lines 195–200 as follows:

*"The natural (physical) and socio-economic indicators of flood risk were analysed using Arc-GIS and remote sensing techniques. The output layer of corresponding indicators (Level-1) and sub indicators (Level-2) were converted into equal pixel-sized raster datasets to map spatial variability of hazard, vulnerability, and finally flood risk index. The indicators are typically selected based on the specific environmental, topographical, socio-economic characteristics and data availability of the study area, allowing for a more accurate representation of regional flood dynamics (Kablan et al., 2017; Meyer et al., 2009). These indicators and sub indicators were then conditioned and rated using inputs from literature and decision-makers, including area experts, academia, and local governments (Table S7) (Danumah et al., 2016; Saaty, 1980, 1988; Stefanidis and Stathis,*

*2013). The conditioning was further validated and corrected for bias using PCA (Ajtai et al., 2023; Maćkiewicz and Ratajczak, 1993). Subsequently, the AHP (Saaty, 1980) was applied to refine and enhance the accuracy of the flood hazard and vulnerability maps".*

Calculations of sub-indicators are provided in the Supplementary Tables (Please refer to supplementary tables S1, S2, S4 and S5). We have also clarified the methodological steps where possible without exceeding the journal's page limitations. All suggested points have been revised and addressed to the best extent possible within the scope of the paper.

**Results:**

**Comments**: Results are presented descriptively, mostly as a narrative to supplement the maps. The PCA-based ranking of indicators (Table 6) is valuable, but the explanation of what the component loadings represent, their physical interpretation, and thresholds for significance are insufficient. The flood hazard and vulnerability maps (Figures 3-5) are critical outputs. However, their validation is either missing or insufficiently described. The risk zones are identified with natural break classification, which is appropriate, but the criteria for defining "High" and "Very High" zones should be clarified. Findings of spatial distributions for different subbasins and provinces are reported, but little explanation exists for why these areas have a higher level of risk beyond cursory understanding. Also, the provided maps have not been designed to publication standards—legends are obscure, font sizes differ, and color choices are hard to distinguish. The authors reference validation from 600 records of flood events, but the comparison still exists only visually, in a qualitative way. Quantitative validation measures such as accuracy, sensitivity, or confusion matrix calculations would add a great deal of strength to findings. There exists no statistical correlation or regression analysis to examine relationships among individual hazard/vulnerability indicators and flood occurrence observations.

**Response:** We have made revisions where possible within the scope and limitations of this study. The PCA ranking of indicators and the interpretation of component loadings are now further clarified in Methodology Section 3.2.1, with a specific focus on the meaning and relevance of each component. Please also refer to lines 285–290, where we describe the classification of risk zones, including the rationale behind the "High" and "Very High" categories.

While quantitative validation falls outside the immediate scope of this research, we acknowledge its importance and appreciate the suggestion. We plan to incorporate statistical measures such as accuracy assessments, confusion matrices, and regression analyses in future work to strengthen model validation.

Regarding the maps (Figures 3–5), we acknowledge the visual quality issues and will provide high-resolution, publication-standard versions in the final manuscript. These will include improved legends, consistent font sizes, and optimized color schemes to enhance interpretability.

**Discussion**

**Comments:** There is little critical examination of the findings. The authors reiterate largely descriptive findings instead of interpreting them more substantively or comparing them with other regional/world studies. The claim that "this study provides a valuable framework" should be tempered with discussion of its limitations, especially regarding validation and uncertainty. The implications for policy or flood management are briefly mentioned but need elaboration. The impact of land cover transformation, infrastructure growth, and institutional capacity towards flood vulnerability is cited, though these factors haven't been substantiated with concrete data or strict analysis. The limitations of the data (e.g., time resolution, aged census data) and methodology (e.g., assumptions in PCA or AHP) aren't addressed. The discussion also fails to include policy-relevant observations specific to Afghanistan's institutional or geographical context.

**Response:** We appreciate the reviewer's thoughtful suggestions. In the discussion section, we have made a more critical analysis of the findings by identifying contributing factors and comparing them with results from other regional and international studies. Furthermore, we have addressed key limitations and uncertainties of the study, particularly those related to data availability and spatial resolution. These aspects are discussed in detail in lines 530–560 of the manuscript. While this study is not intended as a policy paper, it aims to provide insightful information that can support policymakers in understanding spatial flood risk dynamics. A detailed quantitative policy impact assessment was beyond the scope of this work.

**Conclusion**

**Comments:** The conclusion section is generic and lacks analytical depth. Statements are broad and not directly tied to the study's specific findings. The authors should summarize the main results using quantitative evidence and outline specific recommendations based on the risk map. The conclusion should also acknowledge the limitations of the study and suggest directions for future research.

**Response:** In response to the comments, we have outlined the study's limitations and future research directions in a dedicated subsection (5.1: Factors Affecting Flood Risk Assessment and Future Directions). Furthermore, specific recommendations and key challenges derived from our

findings are now elaborated in lines 612–629. While the current study emphasizes a spatial and indicator-based flood risk assessment, we acknowledge the importance of incorporating more quantitative evaluations and will consider this in future work.